# Unified Wisdom: Harnessing Collaborative Learning to Improve Efficacy of Knowledge Distillation

**Atharva Abhijit Tambat**$^*$     *atharvatambat@cse.iitb.ac.in*
*Department of Computer Science and Engineering*
*Indian Institute of Technology, Bombay*

**Durga Sivasubramanian**$^*$     *durgas@cse.iitb.ac.in*
*Department of Computer Science and Engineering*
*Indian Institute of Technology, Bombay*

**Ganesh Ramakrishnan**     *ganesh@cse.iitb.ac.in*
*Department of Computer Science and Engineering*
*Indian Institute of Technology, Bombay*

**Pradeep Shenoy**     *shenoypradeep@google.com*
*Google DeepMind*

**Reviewed on OpenReview:** *https://openreview.net/forum?id=Zj9bb8aQNg*

## Abstract

Knowledge distillation (KD), which involves training a smaller *student* model to approximate the predictions of a larger *teacher* model is useful in striking a balance between model accuracy and computational constraints. However, KD has been found to be ineffective when the teacher and student models have a significant capacity gap. In this work, we address this issue via "meta-collaborative distillation" (MC-DISTIL), where students of varying capacities collaborate during distillation. Using a "coordinator" network (C-NET), MC-DISTIL enables mutual learning among students as a meta-learning task. Our insight is that C-NET learns from each student's performance and training instance characteristics, allowing students of different capacities to improve together. Our method enhances student accuracy for all students, surpassing state-of-the-art baselines, including multi-step distillation, consensus enforcement, and teacher re-training. We achieve average gains of 2.5% on CIFAR-100 and 2% on Tiny ImageNet datasets, consistently across diverse student sizes, teacher sizes, and architectures. Notably, larger students benefiting through meta-collaboration with smaller students is a novel idea. MC-DISTIL excels in training superior student models under real-world conditions such as label noise and domain adaptation. Our approach also yields consistent improvements on the MS COCO object detection benchmark and introduces only a modest 5% computational overhead during training, with no additional cost at inference.

## 1 Introduction

Although modern deep learning methods can effectively learn from large datasets with high-dimensional features, their associated compute and memory requirements often exceed the constraints faced by practical applications. To address this, Knowledge distillation (KD (Hinton et al., 2015)) trains a smaller student model to approximate the higher quality predictions of a larger teacher model. These distilled student models often outperform supervised models of similar capacity; however, such gains are limited by the choice of teacher and student. In particular, larger teacher-student capacity gaps are observed to produce poorer quality student models (Cho & Hariharan, 2019).

---

$^*$Equal contribution.

Previous work on this challenge use an altered teacher (Cho & Hariharan, 2019; Li & Jin, 2022) or introduce additional "assistant" models of intermediate learning capacities between teacher and student model (Mirzadeh et al., 2020; Son et al., 2021). An interesting idea in this literature is the use of multiple teachers for a given student, providing the student with a multi-grain view of the relationship between instance input and label. However, the use of inferior or intermediate-quality teacher models in these approaches potentially limit the quality of information available to the student. We believe, instead, that students should have access to high-quality teacher signals, but focus more on instances that are "learnable" given their model capacity (see e.g., Mindermann et al. (2022) for supervised learning). To this, we add the following insight: we could obtain a finer, multi-grain picture of instance hardness by observing the performance of *students of different capacities* (as opposed to generating and utilizing multiple teachers), which could then be used for better distillation. These observations lead us to the following **problem statement:** Can students of different capacities learn from each other in a collaborative distillation framework, thereby understanding how to prioritize relevant "learnable" information and improving distillation outcomes?

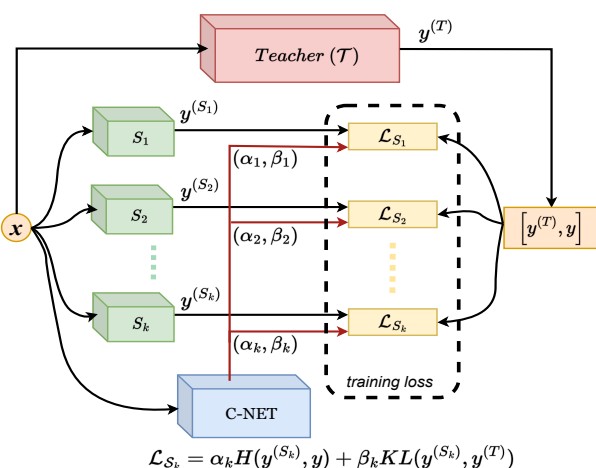

$$\mathcal{L}_{\mathcal{S}_k} = \alpha_k H(y^{(S_k)}, y) + \beta_k KL(y^{(S_k)}, y^{(T)})$$

Figure 1: MC-DISTIL Training Workflow. Losses are weighted according to the parameters obtained from C-NET which are then used to train the individual students.

**Approach:** We build MC-DISTIL – *meta-collaborative distillation* – where students of different capacities are codistilled from a single teacher. We propose to use an auxiliary meta-network for aggregating information across students, and using it to modulate each student's learning on a per-instance basis. We term this meta-learning approach towards joint multi-student distillation as meta-collaboration. As shown in Figure 1, the auxiliary network C-NET modulates the training loss of each student through an instance-dependent reweighting of the teacher and cross-entropy losses. C-NET removes the need for intermediate teachers, and students have access to high-quality teacher signals that they can emulate on a per-instance basis. We set C-NET's objective as the pooled student accuracy on a separate, held-out dataset, setting up a nested optimization that we approximate using an efficient alternating update algorithm (see Section 3 for details). A key advantage of MC-DISTIL is that it produces a spectrum of student models with varying capacities all with improved generalization capabilities, which can be deployed contextually at test time to suit specific application requirements. *Furthermore, the effectiveness of C-Net strengthens as more students participate in the distillation process, enabling richer collaborative gains. As we demonstrate, this framework not only benefits from pooling knowledge across diverse students but also from a fine-grained, adaptive modulation of the distillation process, where training signals are dynamically tailored to both the capacity of each student and the difficulty of individual instances.*

**Pooling information across students:** Figure 2 shows that adding more students consistently improves performance over vanilla KD, as C-NET leverages diverse capacities to better model instance difficulty. We use subsets of ResNet10-xxs, ResNet10-xs, ResNet10-s, and ResNet10-m (He et al., 2016; Kag et al., 2023). For ResNet10-xxs, higher-capacity models are added incrementally, while for ResNet10-m, smaller ones are introduced. Experiments are conducted on the CIFAR-100 dataset (Krizhevsky, 2009) with ResNet10 and ResNet18 as teachers (Figure 2a, 2b). Complete results are provided in Section 4.7.

**Modulating distillation at student & instance granularity:** Figure 3 shows the relative weight assigned in MC-DISTIL to training instances as a function of instance hardness ($x$-axis) and student capacity (different curves). We use the label likelihood gap between teacher and largest student as instance hardness ($x$-axis) and the total weight on loss components ($\alpha + \beta$) as instance weight ($y$-axis). Experiment details are as above.

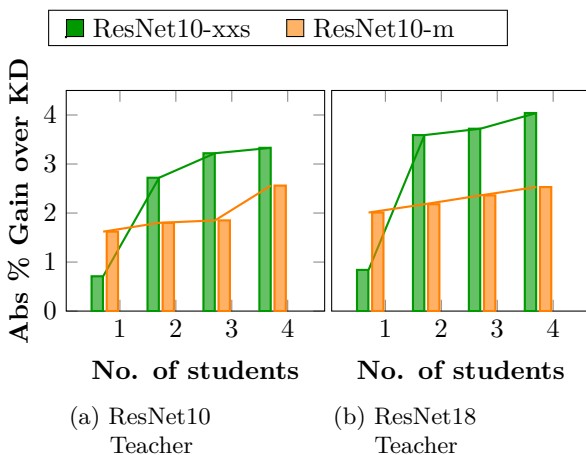

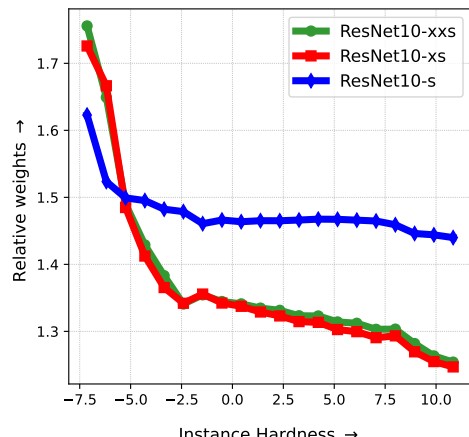

Figure 2: Influence of student cohort size on distillation gains using meta-collaboration. See Section 4.7 for details.

Figure 3: C-NET prioritizes training data based on student capacity and instance hardness. See Section 4.5 for details.

As seen in Figure 3, C-NET implicitly learns instance hardness, encouraging weaker students to focus on easier examples while stronger students learn from a broader range of data. Since instance hardness isn't an explicit input, C-NET learns an implicit mapping of instance difficulty. We further inspect how the choice of loss component weights $\alpha$ and $\beta$ (see Equation 2) helps MC-DISTIL improve KD in Section 4.5.

We extensively evaluate MC-DISTIL across diverse student–teacher architectures and model sizes (Section 4.1.3), demonstrating consistent gains over SOTA benchmarks. MC-DISTIL improves the performance of each student, highlighting the interesting finding that smaller student models can help improve larger students in collaborative distillation. Finally, MC-DISTIL yields improved generalization and robustness (Section 4.4).

## 2 Related works

**Knowledge Distillation (KD)** In supervised learning, Knowledge Distillation (KD) (Hinton et al., 2015) is a valuable method where a 'student' model learns by mimicking a pre-trained 'teacher' model, rather than solely relying on labeled data. The success of KD depends on factors like teacher model accuracy and student model capacity (Menon et al., 2021). Recent research (Harutyunyan et al., 2023) explores its efficacy in relation to supervision complexity. Using early-stopped teacher models has shown promise in improving student training (Cho & Hariharan, 2019), though it requires iterative distillation. Another approach (Liu et al., 2020) employs multiple teacher networks with intermediate knowledge transfer. The strategic blending of loss components (Sivasubramanian et al., 2023) has recently improved KD, particularly in scenarios with significant representation gaps between teacher and student models. The challenge of suboptimal KD performance caused by significant capacity disparities between student and teacher models was addressed by strategically choosing points to learn from the teacher model in Kag et al. (2023). Other innovations such as 'Teacher Assistants' (TAs) or intermediate models have been introduced (Mirzadeh et al., 2020), with further enhancements achieved through stochastic techniques such as Dense Gradient Knowledge Distillation (DGKD) (Son et al., 2021), which involves the simultaneous training of intermediates with occasional model dropout. Recently Li & Jin (2022) showed that the gap between online and offline KD could be bridged using a shadow head on top of the backbone of the teacher model. The shadow head is trained to perform bidirectional distillation. *These works indicate knowledge transfer to the smaller models, resulting in their improvement by the presence of the intermediate bigger models; however the generalization of the larger models has also been shown to improve from the knowledge of smaller models (Mindermann et al., 2022) and bidirectional learning to improve KD (Li & Jin, 2022). Therefore, inspired by these bidirectional signals, we present an approach to train multiple student models simultaneously and communicate vital information via a coordinator network.*

**Online knowledge distillation:** In the absence of a strong pre-trained teacher, online knowledge distillation often involves collaborative model efforts to acquire soft labels. Chen et al. (2020) aggregate predictions from peer models using an attention-based mechanism to obtain soft targets. Wu & Gong (2021) employ an $m$-branch model treating each branch as a peer model, constructing soft distillation targets from weighted logits of these branches and an exponentially moving-averaged $m$-branch model. Guo et al. (2020) propose using pooled student logits with varying learning capacities as soft distillation targets. Additionally, Du et al. (2023) create a curriculum using variance in predictions of multiple students with varying sparsity levels as a measure of task complexity. Zhang et al. (2018) use KL-divergence loss between every student pair to improve consensus among student models. *The primary focus in these works lies either in constructing logits due to the absence of a pre-trained teacher model or explicit synchronization of the student models. In contrast, our approach centers on influencing the learning patterns of the models participating in the collaborative process via signals from peer models.*

**Instance-Specific and Meta-Learning:** Instance-specific learning has been greatly explored to improve training in the presence of noise. Saxena et al. (2019); Algan & Ulusoy (2021); Vyas et al. (2020) propose to learn instance-specific temperature or smoothing parameters to account for potential label noise. In the context of knowledge distillation, Zhao et al. (2021) have demonstrated the advantages of learning instance-level sequences (or curricula) for training samples. A similar instance-wise weighing scheme has been proposed to improve distillation in semi-supervised settings (Iliopoulos et al., 2022). Recent contributions such as that in Ren et al. (2018); Shu et al. (2019); Raghu et al. (2020); Jain & Shenoy (2022) employ meta-learning based on validation sets to acquire instance-specific weights, enhancing robustness. *We introduce a novel approach, MC-DISTIL, that involves the utilization of meta-learning based on validation sets to facilitate a collaborative learning process among multiple models. To the best of our knowledge, this has not been previously explored.*

## 3  MC-Distil: Meta-collaborative distillation

### 3.1  Standard Knowledge Distillation

In supervised learning, one is concerned with training a classifier $y = f_\theta(x)$ using data $\mathcal{D} = \{(x_i, y_i) \mid i \in (1, \cdots, n)\}$. Here, $(x_i, y_i) \in \mathcal{X} \times \mathcal{Y}$ denote pairings of inputs $x_i$ along with their corresponding labels $y_i$ and $\theta$ denote the model parameters. Typically, the labels $y$ are categorical, and practitioners have found that such labels are often inadequate in capturing nuances in the data. A popular mitigation is to incorporate more nuanced 'soft labels', or distributions over labels, as the target for supervision instead of categorical labels. In particular, Knowledge distillation (KD) (Hinton et al., 2015) uses the logits from a pre-trained model (the 'teacher' model) as soft labels for training a classifier, in addition to the standard categorical labels.

Suppose a pre-trained model (teacher) generates logits denoted by $y^{(T)} = \mathcal{T}(\boldsymbol{x})$. Then a new model (student) could be trained using a "teacher matching" objective that involves minimizing the KL-divergence between $y^{(T)}$ and the student's logits $y^{(S)}$. The KD objective is as follows:

$$\mathcal{L}_s = \sum_{\mathcal{D}} \left( (1-\lambda)l_{ce} + \lambda\big(\tau^2 KL\big(y^{(S)}, y^{(T)}\big)\big) \right) \tag{1}$$

Here, $l_{ce} = H\big(y^{(S)}, y\big)$ is a supervised learning loss matching $y^{(S)}$ to the true labels $y$, and $l_{kd} = \tau^2 KL\big(y^{(S)}, y^{(T)}\big)$ is the teacher-matching loss. Typically, $H$ is the standard cross-entropy loss. The hyperparameters $\tau, \lambda$ control the softening of the KL-divergence term, and the relative contributions of the two loss components.

### 3.2  MC-Distil: Multi Student Knowledge Distillation

Successful application of KD often depends on the quality of the pre-trained teacher model (see *e.g.*, (Menon et al., 2021)), and the representational gap between the teacher and student models. To address the capacity gap issue, often several Teacher Assistants (TA) or intermediate models are introduced (Mirzadeh et al., 2020; Son et al., 2021). The success of the such multi-teacher approaches suggests that supervisory inputs from

teacher models of different sizes enrich the information available to the student. We leverage this insight in a completely different setup where a single teacher is simultaneously distilled into multiple cooperating student models. Our primary thesis is that the diversity in student capacities and learning behaviors can be leveraged to tailor the distillation process more effectively for each student.

Specifically, we propose to learn adaptive, instance-wise distillation losses using a coordinator network, denoted as $g_\phi$ (referred to as C-NET). This network modulates the contribution of each training sample to the student models based on their individual learning dynamics and their relationship with the teacher. Formally, we define a set of student models $\mathcal{S} = \{S_j \mid j \in (1, \cdots, k)\}$, a single pre-trained teacher model $\mathcal{T}$, and a shared coordinator network $g_\phi$. The overall training objective for MC-DISTIL is then defined as:

$$\underset{\phi}{\text{argmin}} \sum_{j=1}^{k} \overbrace{\mathcal{L}_{\text{C-NET}}\big( \underbrace{\underset{\theta_j}{\text{argmin}}\, \mathcal{L}_{s_j}(\theta_j, \alpha_j, \beta_j)}_{inner-level}, \mathcal{V}\big)}^{outer-level} \tag{2}$$

$$\text{where } \langle \alpha_{1i}, \ldots, \alpha_{ki}\rangle, \langle \beta_{1i}, \ldots, \beta_{ki}\rangle = g_\phi(x_i), \alpha_j = \langle \alpha_{j1}, \cdots, \alpha_{jn}\rangle, \beta_j = \langle \beta_{j1}, \cdots, \beta_{jn}\rangle$$

$$\text{and } \mathcal{L}_{s_j} = \sum_{(x_i, y_i) \in D} \left( \alpha_{ji} H(x_i, y_i) + \beta_{ji} \tau^2 KL\big(y_i^{(S)}, y_i^{(T)}\big)\right)$$

### 3.2.1 Coordinator network $g_\phi$ (C-Net):

Our key innovation in Equation 2 is to use an additional "coordinator network", which we call C-NET (Figure 1), to address two key challenges, *viz.*, a) reducing the number of learnable parameters and b) introducing interaction among the student models. The C-NET is a learnt function $(A_i, B_i) = g_\phi(x_i)$, where $A_i = [\alpha_{1i}, \ldots, \alpha_{ki}], B_i = [\beta_{1i}, \ldots, \beta_{ki}]$ represent the loss mixing parameters for the $k$ students on the input $x_i$. Therefore, the combined $(A, B)$ for the entire dataset would result in a matrix of size $n \times k$, which can incur a significant memory overhead as the training dataset grows. Thus, to compactly represent the loss mixing parameters we use C-NET parameterized by $\phi \in \Phi$. Moreover, as the C-NET is a centralized mechanism, it acts as a shared interface across students and training instances, enabling the coordinator to incorporate the relative learning dynamics of models with different capacities. This allows C-NET to assign more informed, adaptive weights to each student based on instance-specific difficulty and student model behavior. The training procedure for the C-NET is described in the following section.

**Training the C-Net:** We train the $g_\phi$ (C-NET) using a separate validation set of data $\mathcal{V} = \{(x_i^v, y_i^v) \mid i \in (1, \cdots, m)\}$. The objective is to learn $\phi$ such that the resulting student models generalize well—i.e., they achieve low prediction error on the validation data. Specifically, we minimize the average cross-entropy loss across all $k$ student models on the validation set:

$$\mathcal{L}_{\text{C-NET}} = \frac{1}{km} \sum_{j=1}^{k} \sum_{i=1}^{m} H\big(y_i^{v(S_j)}, y_i^v\big) = \frac{1}{km} \sum_{j=1}^{k} \sum_{i=1}^{m} H\big(f_{\theta_j}(x_i^v), y_i^v\big) \tag{3}$$

Here, $H(\cdot, \cdot)$ denotes the cross-entropy loss between the predicted label from student $j$ and the ground truth. Crucially, $\mathcal{L}_{\text{C-NET}}$ depends on $\phi$ only implicitly — via the student parameters $\theta_j$, which themselves are optimized using the loss mixing weights generated by $g_\phi(\cdot)$ (see Equation 2). In other words, our proposal defines a *bi-level optimization* that encompasses both the C-NET and classifier parameters. Thus, the optimal $\theta$ values depend on the optimal choice of $\phi$ and vice versa.

For most modern deep learning models, solving the inner optimization in Equation 2 in closed form is intractable due to the non-convexity and scale of the models. To address this, we adopt an iterative strategy to approximate the solution to the bilevel optimization problem. Rather than fully optimizing the inner

---

**Algorithm 1** The MC-Distil approach: learning student $\mathcal{S}_1, \cdots \mathcal{S}_k$, Training data $\mathcal{D}$, Validation data $\mathcal{V}$, teacher $\mathcal{T}$ and C-Net $g_\phi$.

---

**Hyperparameters:** $\tau$ Temperature, $\eta_1^1 \cdots \eta_1^k$: learning rates for $k$ students, $\eta_2$ learning rate for C-Net, $L$ epoch interval for C-Net update

---

1: Initialize student model parameters with $\theta_1^0 \cdots \theta_k^0$ and C-Net with $g_\phi^0$
2: **for** $t \in \{0, \ldots, T\}$ **do**
3:     **for** $j \in \{1, \ldots, k\}$ **do**
4:         Update $\theta_j^{t+1}$ by Equation 4.
5:     **end for**
6:     **if** $t\%L == 0$ **then**
7:         $\{x^v, y^v\} \leftarrow \texttt{SampleMiniBatch}(\mathcal{V})$
8:         Compute $\mathcal{L}_{\text{C-Net}}$ using $\{x^v, y^v\}$ and $(\theta_j^{t+1})$ as described in Equation 3.
9:         Update $\phi^{\lfloor \frac{t}{L} \rfloor + 1}$ by Eq. Equation 5.
10:     **end if**
11: **end for**

---

objective before updating the outer parameters, we employ an alternating stochastic gradient descent (SGD) approach. This involves interleaving updates of the student model parameters and the C-Net. Specifically, we perform a few gradient steps on the student parameters using the training set and the weights predicted by the C-Net, followed by an update of the C-Net parameters based on validation loss (Equation 5). This alternating scheme provides a practical and efficient means of optimizing the overall objective. The update steps are summarized as follows:

$$\theta_j^{t+1} = \theta_j^t - \frac{\eta_1^j}{n} \sum_{i=1}^n g_{\phi^t i}^\alpha[j] * \nabla_{\theta_j^t} H\big(y_i^{(S_j)}, y_i\big) + g_{\phi^t i}^\beta[j] * \tau^2 * \nabla_{\theta_j^t} KL\big(y_i^{(S_j)}, y_i^{(T)}\big) \quad \forall j \in \{1, \cdots, k\} \tag{4}$$

$$\phi^{t+1} = \phi^t - \frac{\eta_2}{m} \sum_{i=1}^m \nabla_{\phi^t} \mathcal{L}_{\text{C-Net}}(x_i^v, y_i^v, \Theta^{t+1}) = \phi^t - \frac{\eta_2}{km} \sum_{j=1}^k \sum_{i=1}^m \nabla_{\theta_j^{t+1}} \mathcal{L}_{\text{C-Net}}(x_i^v, y_i^v, \Theta^{t+1}) * \nabla_{\phi^t} \theta_j^{t+1} \tag{5}$$

Here, we denote $\Theta^{t+1} = \langle \theta^{t+1}1, \cdots, \theta^{t+1}k \rangle$ as the collection of updated student model parameters. The learning rates for the student models and the C-Net are represented by $\eta_1^1, \cdots, \eta_1^k$ and $\eta_2$, respectively. We use the notations $g_{\phi^t i}^\alpha[j] = \alpha_j^i$ and $g_{\phi^t i}^\beta[j] = \beta_j^i$ to remind our readers that $\alpha$s and $\beta$s are output from the C-Net ($g(.)$) parameterised by $\phi$. The update step for the C-Net is similar to the standard meta-learning objectives as it uses the updated student model parameters. We present the complete algorithm of MC-Distil in Algorithm 1. Since training C-Net adds to the cost of training, we propose to update C-Net only after $L$ epochs. In Appendix A, we show theoretically that our method converges to the optima of both the validation and training loss functions under some mild conditions.

# 4 Experiments

We conduct experiments on a variety of tasks. First, we compare our method with other knowledge distillation baselines in terms of image classification. We experiment with different settings varying architecture and datasets. Also, we apply our method to the object detection task.

## 4.1 Image Classification

For image classification, we evaluate across diverse datasets—**CIFAR-100** (Krizhevsky, 2009), **Tiny ImageNet** (Le & Yang, 2015) and **ImageNet-1K** (Russakovsky et al., 2015). In addition, we consider **iWildCam** (Beery et al., 2020), **Tiny ImageNet-C** (Hendrycks & Dietterich, 2019), and **Clothing-1M** (Xiao et al., 2015) as part of our further analysis. Dataset details, including splits, input dimensions, and augmentations, appear in Appendix B.1, with training configurations in Appendix B.3.

| CIFAR-100 Test Accuracies | | | | | | | | | | |
|---|---|---|---|---|---|---|---|---|---|---|
| Teacher | | Student | CE | KD | TAKD | DGKD | RMC | DML | SHAKE | MetaDistil | Ours |
| ResNet10-l | 72.20 | ResNet10-xxs | 31.85 | 33.45 | 34.39 | 35.34 | 34.07 | 33.41 | 33.56 | 34.92 | **36.19** |
| | | ResNet10-xs | 42.75 | 44.87 | 44.97 | 47.11 | 45.18 | 44.16 | 42.12 | 46.01 | **47.57** |
| | | ResNet10-s | 52.48 | 55.38 | 56.16 | 57.02 | 53.74 | 55.13 | 56.50 | 57.20 | **58.36** |
| | | ResNet10-m | 64.28 | 66.93 | - | - | 66.66 | 66.06 | 68.80 | 68.28 | **69.42** |
| ResNet10 | 75.18 | ResNet10-xxs | 31.85 | 33.95 | 34.98 | 34.85 | 33.64 | 33.54 | 33.70 | 34.66 | **35.97** |
| | | ResNet10-xs | 42.75 | 44.87 | 45.64 | 46.68 | 42.45 | 44.70 | 44.20 | 46.32 | **47.53** |
| | | ResNet10-s | 52.48 | 55.56 | 56.51 | 56.84 | 53.64 | 55.29 | 56.59 | 57.78 | **58.10** |
| | | ResNet10-m | 64.28 | 67.27 | - | - | 66.58 | 66.25 | 68.63 | 68.89 | **69.21** |
| ResNet18 | 76.99 | ResNet10-xxs | 31.85 | 33.56 | 34.26 | 34.26 | 33.77 | 34.02 | 32.32 | 34.40 | **35.94** |
| | | ResNet10-xs | 42.75 | 45.02 | 45.27 | 47.33 | 45.14 | 44.10 | 43.82 | 46.24 | **46.99** |
| | | ResNet10-s | 52.48 | 55.73 | 55.41 | 56.70 | 54.03 | 54.82 | 56.82 | 57.40 | **57.50** |
| | | ResNet10-m | 64.28 | 66.42 | - | - | 66.04 | 65.94 | 68.12 | 68.43 | **68.45** |
| ResNet34 | 79.47 | ResNet10-xxs | 31.85 | 33.32 | 34.46 | 35.64 | 34.46 | 33.68 | 33.20 | 33.76 | **36.10** |
| | | ResNet10-xs | 42.75 | 44.94 | 45.92 | 47.21 | 42.78 | 44.45 | 45.80 | 46.43 | **47.12** |
| | | ResNet10-s | 52.48 | 54.73 | 56.17 | 57.12 | 53.58 | 55.28 | 56.20 | 56.91 | **57.67** |
| | | ResNet10-m | 64.28 | 66.52 | - | - | 65.58 | 66.88 | 68.96 | 68.09 | **68.24** |
| Tiny ImageNet Test Accuracies | | | | | | | | | | |
| ResNet10-l | 41.25 | ResNet10-xxs | 13.76 | 13.53 | 13.81 | 14.34 | 13.69 | 13.95 | 14.70 | 14.78 | **15.48** |
| | | ResNet10-xs | 18.56 | 19.19 | 19.22 | 20.54 | 19.04 | 19.57 | 19.92 | 20.13 | **21.68** |
| | | ResNet10-s | 24.56 | 25.95 | 26.35 | 27.24 | 25.86 | 26.20 | 27.00 | 27.06 | **29.74** |
| | | ResNet10-m | 33.47 | 34.63 | - | - | 33.72 | 35.34 | 36.66 | 36.02 | **38.26** |
| ResNet10 | 44.04 | ResNet10-xxs | 13.76 | 13.80 | 14.01 | 14.52 | 13.83 | 14.01 | 14.20 | 14.19 | **15.36** |
| | | ResNet10-xs | 18.56 | 19.48 | 19.09 | 21.21 | 19.28 | 19.78 | 19.87 | 20.13 | **21.78** |
| | | ResNet10-s | 24.56 | 26.95 | 25.58 | 26.99 | 26.18 | 27.04 | 27.79 | 27.06 | **29.82** |
| | | ResNet10-m | 33.47 | 35.50 | - | - | 34.78 | 35.79 | 38.01 | 36.02 | **38.54** |
| ResNet18 | 47.94 | ResNet10-xxs | 13.76 | 14.12 | 14.53 | 13.87 | 14.08 | 14.28 | 14.58 | 14.24 | **15.26** |
| | | ResNet10-xs | 18.56 | 19.78 | 19.35 | 19.54 | 19.75 | 18.24 | 19.85 | 19.96 | **21.36** |
| | | ResNet10-s | 24.56 | 26.30 | 26.17 | 27.42 | 25.08 | 25.92 | 27.92 | 27.32 | **30.36** |
| | | ResNet10-m | 33.47 | 35.08 | - | - | 33.37 | 36.77 | 38.60 | 36.08 | **39.11** |
| ResNet34 | 50.10 | ResNet10-xxs | 13.76 | 14.43 | 13.47 | 14.58 | 13.78 | 14.30 | 14.67 | 13.96 | **15.24** |
| | | ResNet10-xs | 18.56 | 19.72 | 18.33 | 20.84 | 19.28 | 19.29 | 19.76 | 20.93 | **21.82** |
| | | ResNet10-s | 24.56 | 27.05 | 24.96 | 27.89 | 25.99 | 26.81 | 28.05 | 27.64 | **29.59** |
| | | ResNet10-m | 33.47 | 35.94 | - | - | 33.58 | 35.88 | 37.97 | 36.88 | **38.34** |

Table 1: Comprehensive comparison of methods across datasets. For each teacher model, we perform knowledge distillation with a group of student models. MC-DISTIL (Ours, last column) substantially improves the average accuracy on unseen test data compared to other distillation baselines, especially those designed for multi-student setups. The highest accuracies are highlighted in bold.

### 4.1.1 Model Architecture and Training

To demonstrate the utility of our method across groups of different model sizes we experiment with a group of ResNet (He et al., 2016) models and several recent larger models. We use the ResNet32 model as C-NET with the classification head changed to output weighting parameters. We present results of varying C-NET size in Section 4.8. We use ResNet10-xxxs, ResNet10-xxs, ResNet10-xs, ResNet10-s and ResNet10-m (Kag et al., 2023) models as the student models. These models are simultaneously trained with either ResNet10-l, ResNet10, ResNet18 or ResNet34 models as a teacher model. In Appendix B.2 we present details of these models. We present experiment results on these combinations in Table 1. Henceforth, in tables, - denotes that for baselines TAKD (Mirzadeh et al., 2020) and DGKD (Son et al., 2021), the initial distillation step coincides with standard KD from the teacher to the largest student, and is therefore not reported separately.

| CIFAR-100 Test Accuracies | | | | | | | | | | | |
|---|---|---|---|---|---|---|---|---|---|---|---|
| Teacher | | Student | CE | KD | TAKD | DGKD | RMC | DML | SHAKE | MetaDistil | Ours |
| ResNet-32x4 | 80.10 | ResNet-8x4 | 71.12 | 72.62 | 74.26 | 74.45 | 73.89 | 73.37 | 73.96 | 73.12 | **75.38** |
| | | MobileNet-V2x2 | 69.52 | 71.78 | 72.07 | 72.27 | 71.23 | 72.46 | 72.17 | 72.53 | **73.05** |
| | | WideResNet-16x2 | 72.79 | 73.34 | 73.72 | 74.12 | 75.19 | 73.87 | 74.22 | 74.10 | **75.52** |
| | | ShuffleNet-V2 | 73.73 | 75.33 | - | - | 76.52 | 75.98 | 76.12 | 76.74 | **77.97** |
| WideResNet-40x2 | 77.60 | ResNet-8x4 | 71.12 | 72.77 | 73.82 | 74.63 | 73.84 | 74.12 | 74.31 | 74.49 | **76.73** |
| | | MobileNet-V2x2 | 69.52 | 72.69 | 72.69 | 71.61 | 70.90 | 73.97 | **74.68** | 74.20 | 74.36 |
| | | WideResNet-40x1 | 72.90 | 73.01 | 73.72 | 74.67 | 75.44 | 74.26 | 74.41 | 75.38 | **75.85** |
| | | ShuffleNet-V2 | 73.73 | 75.85 | - | - | 76.72 | 77.87 | 78.08 | **78.17** | 77.94 |
| Tiny ImageNet Test Accuracies | | | | | | | | | | | |
| ResNet-32x4 | 50.20 | ResNet-8x4 | 37.16 | 37.23 | 38.83 | 39.52 | 36.76 | 38.86 | 38.12 | 40.46 | **41.89** |
| | | MobileNet-V2x2 | 47.68 | 49.89 | 49.89 | 48.21 | 48.07 | 48.24 | 49.79 | 49.85 | **49.95** |
| | | WideResNet-16x2 | 39.11 | 39.47 | 41.66 | 41.77 | 39.77 | 41.35 | 41.98 | 42.12 | **43.88** |
| | | ShuffleNet-V2 | 47.76 | 50.44 | - | - | 49.46 | 49.24 | 49.46 | 50.62 | **52.38** |

Table 2: Comprehensive comparison of methods when training models with larger learning capacity. Here again MC-DISTIL (Ours, last column) substantially improves the test accuracy compared to other distillation baselines even in this setting. The highest accuracies are highlighted in bold.

To illustrate the utility of our method in larger vision models we perform knowledge distillation with ResNet-32x4 as the teacher and ResNet-8x4, ShuffleNet-V2 (Ma et al., 2018), WideResNet-16x2 (Zagoruyko & Komodakis, 2016) and MobileNet-V2x2 (Sandler et al., 2018) as the group of student models. We also perform knowledge distillation with WideResNet-40x2 as a teacher model and ResNet-8x4, ShuffleNet-V2, WideResNet-40x1, and MobileNet-V2x2 as a student model group in the large vision model setting. The results of these experiments are presented in Table 2.

### 4.1.2 Baselines

In our comparative analysis, we assess the performance of our method against a selection of recent works in the field of knowledge distillation, alongside standard knowledge distillation and Empirical Risk Minimization (ERM) or Cross-Entropy based training (CE). Specifically, we compare against TAKD (Mirzadeh et al., 2020) and DGKD (Son et al., 2021) baselines involving multiple students or intermediate models, RMC(Du et al., 2023) uses variance in predictions of students of levels of sparsity as a measure of task complexity for each instance, DML (Zhang et al., 2018) forces collaboration among the models by introducing a KL-divergence loss across different models and SHAKE (Li & Jin, 2022) which introduces a pseudo teacher that allows bidirectional learning during KD. We also compare MC-DISTIL against one more baseline that involves distilling knowledge to each of the students independently using a network architecturally similar to C-NET. We refer to this baseline as the **MetaDistil**. This baseline is similar to AMAL (Sivasubramanian et al., 2023); the strategic mixing loss components are achieved via the C-NET optimization. Further, in instance dependent label-noise setting we compare against L2R (Ren et al., 2018) and MWN (Shu et al., 2019) reweighing techniques created for eliminating label noise by using bi-level optimization and MCD (Gal & Ghahramani, 2016) which uses dropout to model uncertainties. Please refer to Appendix B.4 for more information on these methods.

### 4.1.3 Improving efficacy of Knowledge Distillation

In Table 1, we present results from experiments conducted on CIFAR-100 and Tiny ImageNet datasets, exploring scenarios with a significant capacity gap between teacher and student models. We start with ResNet10-l as the teacher and go on to continue increasing learning capacity of the teacher model and perform knowledge distillation with ResNet10, ResNet18, and ResNet34. On both the datasets, MC-DISTIL, by virtue of meta-collaboration, achieves the best performance among the baselines showing accuracy gains of up to 4% on both the datasets compared to KD. The gains are much more pronounced on the larger models in the student pool for the Tiny ImageNet dataset owing to the increased difficulty in classifying it whereas for

| ImageNet-1K Test Accuracies | | | | | | | |
|---|---|---|---|---|---|---|---|
| Teacher | | Student | CE | KD | DGKD | SHAKE | MetaDistil | MC-Distil |
| ResNet101 | 81.68 | ResNet14 | 22.28 | 19.00 | 20.92 | 21.15 | 22.84 | **23.79** |
| | | ResNet20 | 26.87 | 22.10 | 24.31 | 24.48 | 26.44 | **27.94** |
| | | ResNet32 | 31.04 | 26.83 | 29.34 | 29.26 | 31.56 | **33.07** |
| | | ResNet56 | 37.56 | 33.38 | - | 34.82 | 37.90 | **39.73** |

Table 3: Comprehensive comparison ImageNet-1K test accuracies of student models distilled from a ResNet101 teacher using different knowledge distillation techniques. The proposed method (MC-DISTIL) consistently outperforms baseline approaches across all student architectures. Bold indicates the highest accuracies.

CIFAR-100 the gains are pretty uniform across the student models. These gains are consistent across a wide range of student and teacher capacities. MC-DISTIL improves all of the student models' performances as compared to the baselines, thereby showing that joint distillation of knowledge to a student set is beneficial for both smaller and larger students. We analyze the effect of introducing student models one by one in the presence of C-NET in Section 4.7.

**MC-Distil remains competitive even in scenarios with a small capacity difference.** As illustrated in Table 2, MC-DISTIL maintains its competitive advantage over KD, even when the student model closely matches the size of the teachers, achieving gains of up to 3% relative to KD. These improvements can be attributed to two key factors: (i) *the reweighing of loss terms* and (ii) *meta-collaboration*. The reweighting strategy aligns the learning focus with the model's capacity, emphasizing examples that are more suitable for the student. This targeted weighting helps explain the performance improvement observed with MetaDistil over standard KD, particularly for students like 'ResNet-m'. However, just this reweighing is not sufficient for students such as 'ResNet-xxs' and 'Resnet-s' in the case of larger teachers. This is where MC-DISTIL's ability to leverage C-NET as a communication channel among student models is useful in enhancing knowledge transfer from the teacher model. While the benefits of information flow from intermediate models to smaller ones, as demonstrated in previous studies (Son et al., 2021; Mirzadeh et al., 2020), are well-established for improving final performance, the reverse scenario remains under-explored. The gains reported in Tables 1 and 2 indicate that larger models can also benefit from information exchange with smaller models, akin to standard supervised settings (Mindermann et al., 2022).

**MC-Distil outperforms bidirectional learning methods in KD.** We compare against two KD methods *viz.* SHAKE's (Li & Jin, 2022) and DML (Zhang et al., 2018), both aim to enhance teacher outputs to improve KD's effectiveness. SHAKE learns a pseudo-teacher (a few additional learnable layers on the teacher's backbone) to adapt according to the student model. This performs relatively well for larger students but struggles to accommodate the specific needs of smaller students in Table 1. In contrast, MC-DISTIL carefully controls the knowledge transfer from the teacher model, proving more beneficial for smaller students. On the other hand, DML's forceful explicit collaboration among student cohorts results in inferior performances. This is because aligning outputs restricts larger models from learning to their full capacity and burdens smaller models with the task of matching the outputs of their larger counterparts, similar to traditional KD. In Section 4.10 we show that such collaboration can be added to our setup and if carefully reweighted can lead to improved performances.

### 4.1.4 MC-Distil in extreme teacher student gap settings

To evaluate the resilience of MC-DISTIL under a substantial teacher–student capacity gap, we conduct knowledge distillation on the ImageNet-1K dataset using a high-capacity ResNet101 teacher model, which is significantly larger than the student models ResNet14, ResNet20, ResNet32, and ResNet56. Due to computational constraints, we compare only against the most competitive baselines: DGKD, SHAKE, and MetaDistil. The results for the same are given in Table 3.

Given ImageNet's complexity and the large teacher-student disparity, KD (Knowledge Distillation) under-performs even compared to standard Cross-Entropy (CE) training, aligning with prior findings from Cho & Hariharan (2019). The alternative methods designed to mitigate poor transfer (such as DGKD and SHAKE)

| MS COCO Test Average Precision Values | | | | | | | | | | |
|---|---|---|---|---|---|---|---|---|---|---|
| Metric | Teacher | | Student | CE | KD | TAKD | DGKD | RMC | SHAKE | MetaDistil | Ours |
| **mAP** | RN50 | 46.10 | RN18 | 28.04 | 31.49 | 33.91 | 34.06 | 32.18 | 35.13 | 36.51 | **38.20** |
| | | | RN34 | 31.10 | 37.11 | - | - | 36.06 | 39.79 | 40.69 | **42.70** |
| **AP50** | RN50 | 60.00 | RN18 | 45.45 | 51.12 | 51.45 | 51.73 | 51.44 | 50.91 | 51.70 | **52.90** |
| | | | RN34 | 51.71 | 57.41 | - | - | 56.61 | 58.37 | 58.36 | **58.49** |
| **AP75** | RN50 | 50.20 | RN18 | 28.90 | 32.86 | 34.04 | 35.06 | 36.57 | 37.07 | 39.63 | **41.59** |
| | | | RN34 | 32.34 | 38.78 | - | - | 42.44 | 41.97 | 44.41 | **46.48** |

Table 4: Comparison of mAP, AP50, and AP75 on MS COCO Dataset (Lin et al., 2014) for Faster R-CNN with ResNet50-FPN teacher distilled into ResNet18-FPN and ResNet34-FPN students using various distillation methods. MC-DISTIL (Ours, last column) consistently improves performance across all metrics. Bold indicates the highest accuracies.

provide some improvements over KD, they still fail to surpass CE, with the exception of MetaDistil, which only matches CE performance. Despite these challenges, MC-DISTIL achieves a consistent 1-2% improvement over CE, demonstrating its effectiveness even in extreme teacher-student gap scenarios.

## 4.2 Object Detection

For object detection, we evaluate on the MS COCO benchmark (Lin et al., 2014), using the standard metrics—mAP, AP50, and AP75. The distillation baselines are the same as in the image classification experiments—KD (Hinton et al., 2015), TAKD (Mirzadeh et al., 2020), DGKD (Son et al., 2021), RMC (Du et al., 2023), and SHAKE (Li & Jin, 2022)—with the KD loss adapted to object detection similar to described in Chen et al. (2017). Dataset details, including splits, input dimensions, and augmentations, appear in Appendix B.1, with training configurations in Appendix B.3.

### 4.2.1 Model Architecture and Training

We employ Faster R-CNN (Ren et al., 2015) as the base detector, with a high-capacity ResNet50-FPN teacher distilled into lower-capacity ResNet18-FPN and ResNet34-FPN students. We use the ResNet34 model as C-Net with the classification head changed to output weighting parameters. All models are initialized from ImageNet-pretrained weights and fine-tuned following standard detection pipelines. Appendix B.2 presents details of these models.

### 4.2.2 Boosting Detection Performance with MC-Distil

Table 4 reports detection performance on MS COCO. Across all metrics—mAP, AP50, and AP75—MC-DISTIL achieves the highest performance, surpassing both traditional KD and advanced distillation baselines such as TAKD, DGKD, and RMC. Notably, MC-DISTIL provides an absolute gain of up to 4–5 mAP points over KD, with larger improvements on stricter thresholds like AP75, underscoring its ability to transfer fine-grained localization knowledge effectively. While SHAKE performs well on ResNet34-FPN, it underperforms on the smaller ResNet18-FPN due to its pseudo-teacher bias, whereas MC-DISTIL adapts the teacher's knowledge to both small and mid-sized students through meta-collaboration. The consistent performance boost across all IoU thresholds highlights that our approach better aligns detection outputs between teacher and students, resulting in more precise bounding box predictions.

## 4.3 MC-Distil trains models with better generalisation

We perform knowledge distillation on the challenging iWildCam dataset (Beery et al., 2020). The challenge arises from variations in the capturing environment, resulting in a demanding test set (see Appendix B.1). Table 5 shows the performance of various knowledge distillation methods based on multi-student setups,

| iWildCam Test Accuracies | | | | | | | | |
|---|---|---|---|---|---|---|---|---|
| Teacher | | Student | CE | KD | DGKD | DML | SHAKE | MetaDistil | MC-Distil |
| ResNet10-l | 89.05 | ResNet10-xxs | 75.40 | 75.92 | 76.17 | 76.12 | 76.72 | 76.85 | **76.96** |
| | | ResNet10-xs | 77.30 | 77.37 | 77.86 | 77.50 | 78.30 | 78.27 | **78.82** |
| | | ResNet10-s | 79.97 | 80.42 | 80.30 | 80.51 | 81.79 | 81.94 | **82.80** |
| | | ResNet10-m | 84.77 | 84.94 | - | 85.80 | 86.91 | 86.52 | **87.35** |
| ResNet10 | 91.86 | ResNet10-xxs | 75.40 | 75.85 | 76.20 | 76.17 | 76.25 | 76.82 | **76.94** |
| | | ResNet10-xs | 77.30 | 77.45 | 77.65 | 77.72 | 78.38 | 78.04 | **78.97** |
| | | ResNet10-s | 79.97 | 80.49 | 80.58 | 80.61 | 82.19 | 81.73 | **82.75** |
| | | ResNet10-m | 84.77 | 85.07 | - | 85.28 | 86.34 | 86.44 | **87.49** |

Table 5: Results of knowledge distillation on the challenging iWildCam Dataset (Beery et al., 2020). We compare MC-DISTIL against standard and advanced distillation baselines, observing substantial improvements in test accuracy across all student models. The highest accuracies are highlighted in bold.

| Tiny Imagenet-C (OOD test set) Accuracies | | | | | | | | |
|---|---|---|---|---|---|---|---|---|
| Student | CE | KD | MCD | DGKD | DML | SHAKE | MetaDistil | MC-Distil |
| Corruption level 1 Accuracies | | | | | | | | |
| ResNet10-xxs | 11.25 | 11.38 | 10.61 | 10.77 | 09.99 | 11.47 | 11.02 | **12.05** |
| ResNet10-xs | 14.68 | 15.82 | 14.67 | 15.43 | 15.51 | 16.56 | 16.02 | **16.73** |
| ResNet10-s | 20.88 | 21.97 | 21.94 | 23.18 | 20.29 | 22.58 | 23.97 | **24.88** |
| ResNet10-m | 26.91 | 28.56 | 27.08 | 29.56 | 28.09 | 30.40 | 30.43 | **31.00** |
| Corruption level 2 Accuracies | | | | | | | | |
| ResNet10-xxs | 10.65 | 10.79 | 09.54 | 10.42 | 09.44 | 10.88 | 10.52 | **11.30** |
| ResNet10-xs | 13.75 | 14.60 | 13.72 | 14.60 | 14.56 | 15.42 | 14.78 | **15.52** |
| ResNet10-s | 19.28 | 20.64 | 20.46 | 21.26 | 18.74 | 20.74 | 21.93 | **23.04** |
| ResNet10-m | 24.76 | 25.87 | 24.36 | 26.97 | 26.20 | 27.94 | 27.39 | **28.67** |

Table 6: Comprehensive table presenting an assessment of model (trained with Tiny ImageNet) robustness by conducting inference on the Tiny ImageNet-C Dataset (Hendrycks & Dietterich, 2019) across different degrees of image corruption, alongside results for all baseline models. The columns indicate the method employed for training the model on the Tiny Imagenet Dataset, utilizing ResNet18 as the teacher model. Corruption level 2 reflects a higher severity of corruption than level 1. The highest accuracies are highlighted in bold.

including MC-DISTIL. Across both weak (ResNet10-l) and strong (ResNet10) teachers, and for a wide range of student model capacities–from extremely small (ResNet10-xxs) to moderately sized (ResNet10-m)–MC-DISTIL consistently achieves superior test accuracies.

The consistent performance of MC-DISTIL highlights its robustness in leveraging instance-level importance through soft selection, enabling better generalization in complex, real-world scenarios such as iWildCam. This reinforces the effectiveness of our method not only in standard benchmarks but also under real-world distribution shifts and limited model capacities.

Using models trained with ResNet18 as the teacher on the Tiny ImageNet dataset (reported in Table 1), we classify Tiny ImageNet-C (Hendrycks & Dietterich, 2019). Table 6 presents the accuracies obtained by models trained with different KD methods across two levels of corruption (details in Appendix B.1). In Appendix C.2 we present the complete table with results obtained across all five corruption levels. Notably, MC-DISTIL produces models that demonstrate robustness and superior generalization, even when the test data distribution differs from the training set.

| Student | CE | KD | MCD | MWN | L2R | MetaDistil | MC-Distil |
|---|---|---|---|---|---|---|---|
| **Instance CIFAR-100 Accuracies** | | | | | | | |
| RN10xxxs | 23.69 | 23.67 | 24.33 | 24.14 | 24.88 | 26.48 | **27.07** |
| RN10xxs | 28.97 | 28.55 | 29.81 | 28.66 | 29.04 | 29.59 | **31.88** |
| RN10xs | 38.04 | 38.19 | 38.88 | 37.34 | 39.39 | **42.57** | 42.18 |
| RN10s | 48.40 | 49.22 | 49.71 | 45.84 | 48.14 | 52.67 | **53.26** |
| RN10m | 59.42 | 60.14 | 61.62 | 55.74 | 60.90 | 60.64 | **61.61** |
| **Clothing-1M Accuracies** | | | | | | | |
| RN10xxxs | 43.56 | 44.58 | 45.65 | 41.72 | 45.72 | 46.08 | **46.43** |
| RN10xxs | 51.67 | 51.28 | 52.22 | 48.74 | 51.60 | **52.88** | 52.58 |
| RN10xs | 57.14 | 57.16 | 57.51 | 53.49 | 56.70 | 57.06 | **58.28** |
| RN10s | 61.00 | 62.02 | 61.59 | 48.74 | 60.34 | 61.31 | **62.27** |
| RN10m | 64.83 | 66.55 | 64.86 | 61.21 | 63.45 | 63.64 | **64.92** |

Table 7: Assessment of performances of de-noising and distillation methods on instance-specific noisy datasets. Results are reported for multiple student models distilled from a ResNet18 teacher on Instance CIFAR-100 (Xia et al., 2020) (top) and Clothing-1M (Xiao et al., 2015) (bottom). Our method (MC-DISTIL) consistently achieves the best performance, outperforming 4 out of 5 student models on both datasets.

### 4.4 KD in Instance Dependent Label-Noise

We now evaluate MC-DISTIL under a setting where instance-dependent label noise is present in the labels. We illustrate this experiment on two datasets, namely: *Instance CIFAR-100* (Xia et al., 2020) and *Clothing-1M* (Xiao et al., 2015) where the labels inherit noise due to the data generation process as against randomly flipping the labels. The results are reported in Table 7. Since MC-DISTIL employs a reweighting scheme, C-NET learns weights to reduce the impact of the noise. For the inputs with high noise, it is able to assign low weight to CE loss and rely more heavily on the distillation term, thereby outperforming other methods. Furthermore, the joint distillation (by communication via C-NET) of different-sized student models provides additional valuable information to combat noise effectively.

### 4.5 Why MC-Distil uses instance and loss component specific weights

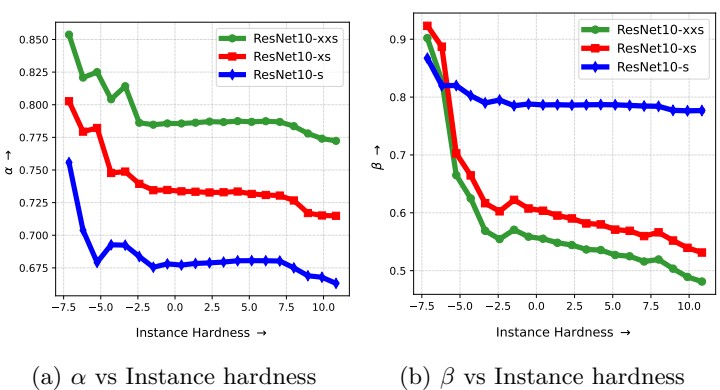

(a) $\alpha$ vs Instance hardness (b) $\beta$ vs Instance hardness

Figure 4: Plot between student model errors and weighting parameters during knowledge distillation is performed using ResNet18 as the teacher model on the CIFAR-100 Dataset (Krizhevsky, 2009). It illustrates how MC-DISTIL adaptively adjusts the loss component weights to guide students of varying capacities

We expand on the analysis presented in Figure 3 to understand how C-NET assigns weights to individual loss components. Figure 4a shows how $\alpha$s, parameter controlling learning from the hard label and Figure 4b shows how $\beta$s, parameter controlling learning from the teacher model (see Equation 2) changes with instance hardness for models with different learning capacity. We define instance hardness as the teacher–largest-student likelihood gap because a larger likelihood gap indicates that the student struggles to match the teacher's confidence, implying higher instance difficulty. For the smaller student models, learning majorly happens via hard labels, whereas larger models prefer learning from the teacher's nuanced soft labels. KD improves the label stu-

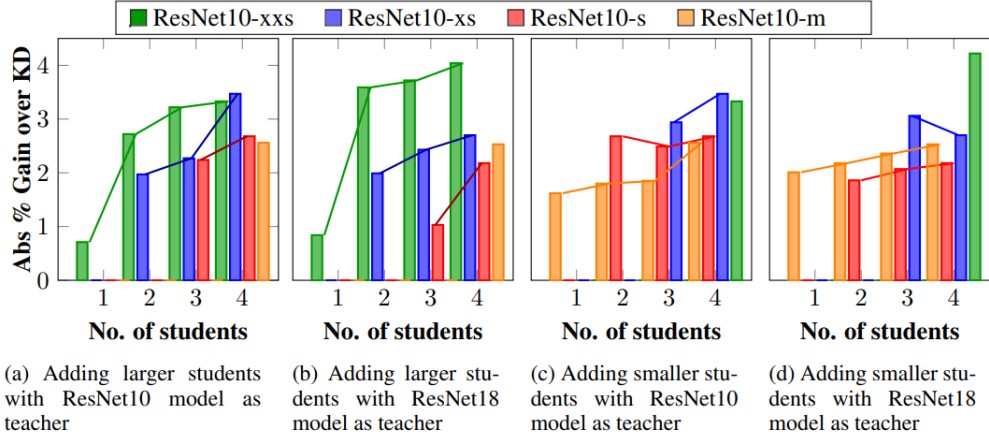

Figure 5: Analysis of the effect of introducing student model cohorts for different learning capacities in meta-collaboration setting.

dent model's performance by offering nuanced soft labels as targets. However, when the student model cannot encode these nuanced soft labels, their introduction can worsen the student model's performance Cho & Hariharan (2019). Therefore, choosing a favorable ground truth is also important along with focusing on the "learnable" points. MC-Distil achieves this by using two sets of weighing parameters–$\alpha$s and $\beta$s, and carefully tuning it according to student model capacity. For the smaller models, MC-Distil carefully introduces soft teacher labels of only the easy points, and thus MC-Distil improves the smaller model's performance over traditional KD.

## 4.6 Scalability of MC-Distil with very large student cohort

| Teacher | | Student | CE | KD | MC-Distil |
|---|---|---|---|---|---|
| ResNet18 | 76.99 | ResNet10-xxs | 31.85 | 33.56 | 35.94 |
| | | ResNet10-xs | 42.75 | 45.02 | 46.99 |
| | | ResNet10-xs | 52.48 | 55.73 | 57.50 |
| | | ResNet10-m | 64.28 | 66.42 | 68.45 |
| | | ResNet10-l | 72.20 | 73.85 | 74.55 |
| | | ResNet10 | 75.18 | 76.73 | 77.20 |

Table 8: Performance of MC-Distil as the number of student cohorts increases on CIFAR-100 Dataset (Krizhevsky, 2009).

As demonstrated in Section 4.1.4 MC-Distil scales effectively to ImageNet-1K, a large-scale dataset with over 1.2 million training images and 1,000 classes. To further demonstrate scalability, we extend this analysis to six student models of increasing capacity on the CIFAR-100 dataset. As shown in the Table 8, MC-Distil consistently outperforms both cross-entropy (CE) training and standard knowledge distillation (KD) across all student sizes — from very compact models like ResNet10-xxs to full-capacity ResNet10. For instance, the smallest model (ResNet10-xxs) improves from 31.85% (CE) to 35.94% (MC-Distil), while even the largest student (ResNet10) benefits, reaching 77.2% accuracy compared to 76.73% with KD. This consistent improvement highlights MC-Distil's robustness across a wide range of model capacities.

## 4.7 Ablation: Changing size of student cohort

To investigate the impact of introducing additional students in the presence of C-Net, we conducted an experiment in which we incrementally introduced student models, one at a time. We present the results of these experiments in Figure 5. This experiment was conducted in two distinct settings: one in which each subsequent addition involved a student with a larger learning capacity. This is presented in Figure 5a and 5b. The other setting is in which each additional introduced student is of a smaller learning capacity as shown in Figure 5c and 5d. In both settings, we perform knowledge distillation with two teachers *viz.*, ResNet10 and ResNet18. We note that across different teachers, introducing additional students improves the performances of all the students participating in the training process. This is much more pronounced in the setting where larger students are added.

| Teacher | | Student | C-Net | | | |
|---|---|---|---|---|---|---|
| | | | ResNet10-l | ResNet10 | ResNet18 | ResNet32 |
| ResNet10 | 75.18 | ResNet10-xxs | 36.13 | 34.95 | 36.00 | 36.38 |
| | | ResNet10-xs | 47.40 | 46.88 | 47.19 | 47.69 |
| | | ResNet10-s | 56.93 | 57.48 | 57.70 | 58.24 |
| | | ResNet10-m | 68.65 | 67.95 | 67.91 | 69.11 |
| ResNet18 | 76.99 | ResNet10-xxs | 36.12 | 34.48 | 35.28 | 36.14 |
| | | ResNet10-xs | 47.68 | 46.44 | 46.98 | 47.68 |
| | | ResNet10-s | 57.12 | 57.82 | 57.65 | 58.21 |
| | | ResNet10-m | 69.72 | 69.72 | 68.96 | 69.62 |
| ResNet34 | 79.47 | ResNet10-xxs | 35.93 | 34.71 | 35.65 | 36.13 |
| | | ResNet10-xs | 47.31 | 46.44 | 46.42 | 47.52 |
| | | ResNet10-s | 57.64 | 57.30 | 56.89 | 57.70 |
| | | ResNet10-m | 69.28 | 68.20 | 67.54 | 68.49 |
| ResNet10-l | 41.25 | ResNet10-xxs | 35.88 | 34.35 | 35.76 | 35.90 |
| | | ResNet10-xs | 47.28 | 45.80 | 46.59 | 47.42 |
| | | ResNet10-s | 57.51 | 56.61 | 56.49 | 57.74 |
| | | ResNet10-m | 68.86 | 67.94 | 67.49 | 68.54 |

Table 9: The table reports test accuracies for various student–teacher–C-Net combinations, analyzing the impact of C-Net size on model performance over CIFAR-100 Dataset.

## 4.8 Ablation: Varying the size of C-Net

Table 9 presents an analysis of test accuracies on the CIFAR-100 dataset for different student architectures under the influence of various sizes of the coordinater network (C-Net). We observe that, in the majority of instances, there are negligible alterations in the test accuracy of student models when it is trained with coordinator networks (C-Net) of varying sizes. The lack of significant changes suggests that the choice of C-Net size has minimal impact on the student model's ability to capture and generalize from the teacher's knowledge. This stability across different C-Net sizes indicates robustness in the knowledge distillation process, emphasizing that the coordination mechanism implemented by C-Net is effective across a range of architectural scales.

## 4.9 Ablation: Changing size of the validation data

| Teacher | | Validation Set Size | | | |
|---|---|---|---|---|---|
| Model | Acc. | Student | 2% | 5% | 10% |
| ResNet18 | 76.99 | ResNet10-xxs | 36.05 | 36.22 | 36.54 |
| | | ResNet10-xs | 47.23 | 47.53 | 47.72 |
| | | ResNet10-s | 57.73 | 57.91 | 58.00 |
| | | ResNet10-m | 68.93 | 68.95 | 69.01 |

Table 10: Performance of MC-Distil under varying validation dataset sizes, highlighting its sensitivity and overall robustness to validation data availability. Knowledge Distillation is performed using ResNet18 as the teacher model on the CIFAR-100 Dataset (Krizhevsky, 2009).

We allocate 10% of the training data as a validation set, as detailed in Appendix B.1. However, to demonstrate that MC-Distil is not heavily dependent on large validation sets, we conduct additional experiments with reduced validation set sizes—specifically 2% and 5% of the training data. The corresponding test accuracies are reported in Table 10, where ResNet18 is used as the teacher model for distillation on the CIFAR-100 dataset. As shown, MC-Distil maintains stable and competitive performance across all student models even with signif-

icantly smaller validation sets. This indicates that the method is robust and can effectively learn meaningful instance-level weights without requiring requiring supervision from large validation corpora. The performance gains reported in Tables 1 and 2 can therefore be achieved with minimal validation overhead, making MC-DISTIL practical and scalable for real-world applications with limited validation data.

### 4.10 Incorporating online distillation techniques in MC-Distil

**Consensus across students:** To further encourage information sharing across students, we propose an additional term in the loss function (Equation 2). This term minimizes the KL divergence between each student's logits and a *consensus* representation of logits across students (a representation we call the "Pooled Student"). For a training instance $(x, y) \in D$ such that the true label is $c$, *i.e.*, $y[c] = 1$, we first define the PooledStudent logits for each label $l$:

$$y^{(PS)}[l] = \begin{cases} \max\big(y^{(S_1)}[l], \cdots y^{(S_k)}[l]\big), & \text{if } l = c \\ \min\big(y^{(S_1)}[l], \cdots y^{(S_k)}[l]\big), & \text{otherwise} \end{cases} \tag{6}$$

This is similar to MinLogit introduced in Guo et al. (2020). Using the PooledStudent logits, the loss function in Equation 2 is updated to:

$$\mathcal{L}_{s_j} = \frac{1}{n} \sum_{i=1}^{n} \alpha_{ij} H\big(y_i^{(S_j)}, y_i\big) + \beta_{ij}\tau^2 KL\big(y_i^{(S_j)}, y_i^{(T)}\big) + \gamma_{ij}\tau^2 KL\big(y_i^{(S_j)}, y_i^{(PS)}\big) \quad \forall j \in \{1, \cdots, k\} \tag{7}$$

| Teacher | Student | MC-Distil | MC-Distil-PS |
|---|---|---|---|
| ResNet10-l | ResNet10-xxs | 36.29 | **36.44** |
| | ResNet10-xs | 47.14 | **47.65** |
| | ResNet10-s | 57.37 | **57.55** |
| | ResNet10-m | 68.47 | **68.51** |
| ResNet10 | ResNet10-xxs | 36.57 | **36.78** |
| | ResNet10-xs | 47.55 | **48.03** |
| | ResNet10-s | 57.59 | **58.20** |
| | ResNet10-m | 69.03 | **69.50** |

Table 11: We present the analysis of the effect of adding a loss component based on the pooled student to MC-DISTIL. Here MC-DISTIL-PS represents the test result obtained with model trained with pooled student-based loss component.

Correspondingly, the C-NET outputs are extended to be $(A, B, \Gamma) = g_\phi(x)$. Putting it all together, MC-DISTIL minimizes the total loss over all students, *i.e.*, $\mathcal{L}_s = \sum_j \mathcal{L}_{s_j}$. We investigate the impact of incorporating a loss component derived from the pooled student, as introduced in Equation 6, in Table 11. The test results labeled MC-DISTIL-PS correspond to the model trained with the pooled student-based loss component. It is noteworthy that integrating this new logit proves beneficial in enhancing the performance gains achieved through meta-collaborative learning when applied in a controlled manner, as suggested in Equation 6. This approach doesn't enforce explicit collaboration among the students; rather, it serves as an additional source of knowledge. Some students, particularly those finding learning directly from the teacher challenging, may leverage this supplementary knowledge, contributing to improved performance.

### 4.11 Computational Cost Analysis of MC-Distil

Compared to baselines such as TAKD (Mirzadeh et al., 2020), DGKD (Son et al., 2021), and DML (Zhang et al., 2018)—which require multiple student models during training—MC-DISTIL introduces an additional C-NET. For a student pool consisting of ResNet10-xxs, ResNet10-xs, ResNet10-s, and ResNet10-m, with ResNet34 as the teacher model and ResNet10-l as the C-NET, the additional overhead is only 5% in terms of total FLOPs. Furthermore, as demonstrated in Section 4.8, the performance of MC-Distil is not highly sensitive to the size of the C-NET, allowing for further optimization of this component. Importantly, both the C-NET and the student cohort are utilized only during training. At inference time, MC-DISTIL requires no ensemble or multi-student framework and thus incurs no additional cost over traditional KD.

| Method | GFLOPs / Batch | TFLOPs / Epoch | Time / Epoch (s) |
|---|---|---|---|
| TAKD | 01.49 | 00.58 | 20.06 |
| DGKD | 04.77 | 01.86 | 24.63 |
| RMC | 04.74 | 01.85 | 12.49 |
| SHAKE | 18.63 | 07.26 | 14.39 |
| MetaDistil | 10.29 | 04.01 | 09.10 |
| **MC-Distil** | 02.47 | 00.97 | 11.90 |

Table 12: Computational cost, reported as GFLOPs per batch and TFLOPs per epoch, along with training time (in seconds) per epoch for training the full student cohort.

Table 12 presents a summary of the computational cost in terms of GFLOPs per batch and TFLOPs per epoch, as well as the time required to train all student models for one epoch. These measurements provide a fair comparison of the total training overhead associated with each approach. Details of the profiling setup and assumptions are provided in Appendix B.6.

As shown in the table, methods such as TAKD and DGKD require the sequential training of multiple models. This leads to an inherent bottleneck in wall-clock time, since the overall duration is the sum of each intermediate stage. In contrast, MC-Distil trains all students simultaneously, avoiding this cumulative delay.

Although SHAKE and MetaDistil achieve per-epoch training times comparable to our method, they incur a larger total computational cost because they require a separate auxiliary model for each student. For instance, SHAKE uses an individual proxy teacher per student, while MetaDistil trains a separate meta-network for re-weighting the loss. When extended to train multiple students, the overhead of these approaches scales linearly with the number of students, making them significantly more expensive in aggregate compared to MC-Distil, which jointly optimizes the student cohort with a shared C-Net.

While RMC is computationally lighter than SHAKE and MetaDistil, it is outperformed by MC-Distil in terms of final model accuracy. Table 1 provides a detailed comparison across CIFAR-100 and Tiny ImageNet datasets, showing consistent gains in accuracy across all student sizes. This illustrates that MC-Distil achieves a more favorable tradeoff between efficiency and effectiveness. Further details about the calculation methodology of the FLOPS are mentioned in Appendix B.6.

## 5 Conclusion

We introduced MC-Distil, a novel knowledge distillation framework grounded in meta-collaboration, or learning-to-collaborate. Unlike conventional distillation techniques that treat each student model independently or enforce rigid alignment with the teacher, MC-Distil facilitates joint learning among a cohort of students via a dedicated coordinator network, C-Net. This network dynamically modulates the flow of gradient signals during training, enabling each student to learn not only from the teacher but also from the training behavior and representations of its peers. As a result, each student model evolves in a collaborative yet personalized fashion, free from hard constraints to mimic either the teacher or other students.

This flexible coordination fosters the emergence of diverse, high-performing student models that are well-suited to a range of deployment needs with varying compute and memory budgets. MC-Distil enables the training of a full student pool in a single pass, reducing overhead without sacrificing quality. The learned diversity also contributes to robustness, allowing practitioners to select from multiple viable models depending on specific task requirements or real-world constraints.

Our extensive evaluations across CIFAR-100, Tiny ImageNet, and MS-COCO benchmarks demonstrate that MC-Distil consistently surpasses both classical and advanced distillation baselines, including KD, TAKD, DGKD, SHAKE, and RMC. Notably, MC-Distil maintains its edge in the presence of data noise and under domain shifts, highlighting its robustness and generalization capabilities. By aligning students with both teacher knowledge and peer insights, MC-Distil paves the way for scalable, adaptable, and more resilient distillation strategies suitable for practical deployment.

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

## Appendix

## A Convergence

In this section, we lay out the complete proof of theorems. Our proof technique is inspired partly by the prior literature (Shu et al., 2019).

**Lemma 1** *Suppose the meta loss function is Lipschitz smooth with constant $L$, and C-NET$(\cdot)$ is differential with a $\delta$-bounded gradient and twice differential with its Hessian bounded by $\mathcal{B}$, and the loss function $\mathcal{L}$ have $\rho$-bounded gradients with respect to training/metadata. Then the gradient of $\phi$ with respect to meta loss is Lipschitz continuous.*

**Proof** From equation Equation 3

$$\mathcal{L}_{\text{C-NET}} = \frac{1}{km}\sum_{j=1}^{k}\sum_{l=1}^{m} H\big(y_l^{v(S_j)}, y_l^v\big) = \frac{1}{km}\sum_{j=1}^{k}\sum_{l=1}^{m} H\big(f_{\theta_j}(x_l^v), y_l^v\big) \tag{8}$$

The gradient of $\phi$ with respect to meta loss can be written as:

$$
\begin{aligned}
\nabla_\phi \mathcal{L}_{\text{C-NET}l} &= \frac{1}{k}\sum_{j=1}^{k} \nabla_{\theta_j} H\big(f_{\theta_j}(x_l^v), y_l^v\big) * \nabla_\phi \theta_j \\
&= \frac{1}{k}\sum_{j=1}^{k} -\frac{\eta_1^j}{n}\sum_{i=1}^{n} \nabla_{\theta_j} H\big(f_{\theta_j}(x_l^v), y_l^v\big) \\
&\quad * \nabla_\phi\bigg(g_{\phi^t}^\alpha[j] * \nabla_{\theta_j^t} H\big(y_i^{(S_j)}, y_i\big) + g_{\phi^t}^\beta[j] * \tau^2 * \nabla_{\theta_j^t} KL\big(y_i^{(S_j)}, y_i^{(T)}\big)\bigg)
\end{aligned}
\tag{9}
$$

Let $G_{li}^H = \nabla_{\theta_j} H\big(f_{\theta_j}(x_l^v), y_l^v\big) * \nabla_{\theta_j^t} H\big(y_i^{(S_j)}, y_i\big)$ and $G_{li}^{KL} = \nabla_{\theta_j} H\big(f_{\theta_j}(x_l^v), y_l^v\big) * \tau^2 * \nabla_{\theta_j^t} KL\big(y_i^{(S_j)}, y_i^{(T)}\big)$. Then again taking gradients on both sides w.r.t $\phi$,

$$\nabla_\phi^2 \mathcal{L}_{\text{C-NET}l} = \frac{1}{k}\sum_{j=1}^{k} -\frac{\eta_1^j}{n}\sum_{i=1}^{n}\bigg[\nabla_\phi G_{li}^H * \nabla_\phi g_\phi^\alpha + G_{li}^H * \nabla_\phi^2 g_\phi^\alpha \nabla_\phi + \nabla_\phi G_{li}^{KL} * \nabla_\phi g_\phi^\beta + G_{li}^{KL} * \nabla_\phi^2 g_\phi^\beta\bigg] \tag{10}$$

For the first term on the right-hand side, we have that,

$$
\begin{aligned}
\Big\|\nabla_\phi G_{li}^H * \nabla_\phi g_\phi^\alpha\Big\| &\leq \delta\Big\|\nabla_\phi \nabla_{\theta_j} H\big(f_{\theta_j}(x_l^v), y_l^v\big) * \nabla_{\theta_j^t} H\big(y_i^{(S_j)}, y_i\big)\Big\| \\
&= \delta\Big\| -\frac{\eta_1^j}{n}\sum_{i=1}^{n} \nabla_{\theta_j}^2 H\big(f_{\theta_j}(x_l^v), y_l^v\big) \\
&\quad * \nabla_\phi\bigg(g_{\phi^t}^\alpha[j] * \nabla_{\theta_j^t} H\big(y_i^{(S_j)}, y_i\big) + g_{\phi^t}^\beta[j] * \tau^2 * \nabla_{\theta_j^t} KL\big(y_i^{(S_j)}, y_i^{(T)}\big)\bigg) * \nabla_{\theta_j^t} H\big(y_i^{(S_j)}, y_i\big)\Big\|
\end{aligned}
\tag{11}
$$

Since $\Big\|\nabla_{\theta_j}^2 H\big(f_{\theta_j}(x_l^v), y_l^v\big)\Big\| \leq L, \Big\|\nabla_{\theta_j^t} H\big(y_i^{(S_j)}, y_i\big)\Big\| \leq \rho, \Big\|\nabla_{\theta_j^t} KL\big(y_i^{(S_j)}, y_i^{(T)}\big)\Big\| \leq \rho, \Big\|\nabla_\phi g_\phi^\alpha\Big\| \leq \delta$ and $\Big\|\nabla_\phi g_{\phi^t}^\beta\Big\| \leq \delta$,

$$
\begin{aligned}
\delta\Big\| -\frac{\eta_1^j}{n}\sum_{i=1}^{n} \nabla_{\theta_j}^2 H\big(f_{\theta_j}(x_l^v), y_l^v\big) * \nabla_\phi\bigg(g_{\phi^t}^\alpha[j] * \nabla_{\theta_j^t} H\big(y_i^{(S_j)}, y_i\big) + g_{\phi^t}^\beta[j] * \tau^2 * \nabla_{\theta_j^t} KL\big(y_i^{(S_j)}, y_i^{(T)}\big)\bigg) \\
* \nabla_{\theta_j^t} H\big(y_i^{(S_j)}, y_i\big)\Big\| \leq \delta\Big\| -\frac{\eta_1^j}{n}\sum_{i=1}^{n} L * \Big(\delta * \rho + \delta * \rho * \tau^2\Big)\Big) * \rho\Big\| \leq \eta_1^{tj} L \rho^2 \delta^2 (1 + \tau^2)
\end{aligned}
\tag{12}
$$

Similarly,

$$\left\|\nabla_\phi G_{li}^{KL} * \nabla_\phi g_\phi^\beta\right\| \le \eta_1^{tj} L \rho^2 \delta^2 (1 + \tau^2) \tag{13}$$

And for the second term on the right-hand side in Equation 10 we have,

$$\|G_{li}^H * \nabla_\phi^2 g_\phi^\alpha \nabla_\phi\| \le \mathcal{B} * \delta^2 \tag{14}$$

Similarly,

$$\|G_{li}^{KL} * \nabla_\phi^2 g_\phi^\beta\| \le \mathcal{B} * \delta^2 \tag{15}$$

Let $\eta_1 = \max\{\eta_1^1, \cdots, \eta_1^k\}$. Thus combining Equations 12, 13, 14 and 15,

$$\|\nabla_\phi^2 \mathcal{L}_{\text{C-NET}_l}\| \le 2 * \eta_1 \rho^2 (L\delta^2(1 + \tau^2) + \mathcal{B}) \tag{16}$$

Define $L_V = 2 * \eta_1 \rho^2 (L\delta^2(1 + \tau^2) + \mathcal{B})$, based on Lagrange mean value theorem, we have:

$$\|\nabla\mathcal{L}_{\text{C-NET}}(\phi_1) - \nabla\mathcal{L}_{\text{C-NET}}(\phi_2)\| \le L_V \|\phi_1 - \phi_2\|, \ \forall \phi_1, \phi_2 \tag{17}$$

$\blacksquare$

**Theorem 1** *Suppose the loss function $\ell$ is Lipschitz smooth with constant $L$, and C-NET$(\cdot)$ is differential with a $\delta$-bounded gradient and twice differential with its Hessian bounded by $\mathcal{B}$, and the loss function $\mathcal{L}$ have $\rho$-bounded gradients concerning training/metadata. Let the learning rates $\eta_1^{1t}, \cdots, \eta_1^{kt}$ satisfies $\eta_1^{jt} = \min\{1, \frac{a}{T}\} \forall j \in \{1, \cdots, k\}$, for some $a > 0$, such that $\frac{a}{T} < 1$, and $\eta_2^t, 1 \le t \le N$ is a monotone descent sequence, $\eta_2^t = \min\{\frac{1}{L}, \frac{c}{\sigma\sqrt{T}}\}$ for some $c > 0$, such that $\frac{\sigma\sqrt{T}}{c} \ge L$ and $\sum_{t=1}^\infty \eta_2^t \le \infty, \sum_{t=1}^\infty (\eta_2^t)^2 \le \infty$. Then MC-DISTIL can achieve $\mathbb{E}[\|\nabla_\phi \mathcal{L}_{\text{C-NET}}(\Theta^t(\phi^t))\|_2^2] \le \epsilon$ in $\mathcal{O}(1/\epsilon^2)$ steps. More specifically,*

$$\min_{0 \le t \le T} \mathbb{E}[\|\nabla_\phi \mathcal{L}_{\text{C-NET}}(\Theta^t(\phi^t))\|_2^2] \le \mathcal{O}(\frac{C}{\sqrt{T}}), \tag{18}$$

*where $C$ is some constant independent of the convergence process, $\sigma$ is the variance of drawing uniformly mini-batch samples at random.*

**Proof**

From Equation 5

$$\phi^{t+1} = \phi^t - \frac{\eta_2^t}{m} \sum_{i=1}^m \nabla_{\phi^t} \mathcal{L}_{\text{C-NET}}(x_i^v, y_i^v, \Theta^{t+1}) = \phi^t - \eta_2^t \nabla_\phi \mathcal{L}_{\text{C-NET}}(\Theta^{t+1}(\phi^t)) \tag{19}$$

This can be written as,

$$\phi^{t+1} = \phi^t - \eta_2^t \nabla_\phi \mathcal{L}_{\text{C-NET}}(\Theta^{t+1}(\phi^t))\big|_{\Xi_t} \tag{20}$$

Since the mini-batch $\Xi_t$ is drawn uniformly from the entire data set, we can rewrite the update equation as:

$$\phi^{t+1} = \phi^t - \eta_2^t [\nabla_\phi \mathcal{L}_{\text{C-NET}}(\Theta^{t+1}(\phi^t)) + \xi^{(t)}] \tag{21}$$

where $\xi^{(t)} = \nabla_\phi \mathcal{L}_{\text{C-NET}}(\Theta^{t+1}(\phi^t))\big|_{\Xi_t} - \nabla_\phi \mathcal{L}_{\text{C-NET}}(\Theta^{t+1}(\phi^t))$. Note that $\xi^{(t)}$ are i.i.d random variables with finite variance since $\Xi_t$ are drawn *i.i.d* with a finite number of samples. Furthermore, $\mathbb{E}[\xi^{(t)}] = 0$, since samples are drawn uniformly at random. Observe that

$$\mathcal{L}_{\text{C-NET}}(\Theta^{t+2}(\phi^{t+1})) - \mathcal{L}_{\text{C-NET}}(\Theta^{t+1}(\phi^t)) = \left[\mathcal{L}_{\text{C-NET}}(\Theta^{t+2}(\phi^{t+1})) - \mathcal{L}_{\text{C-NET}}(\Theta^{t+1}(\phi^{t+1}))\right]$$

$$+ \left[ \mathcal{L}_{\text{C-Net}}(\Theta^{t+1}(\phi^{t+1})) - \mathcal{L}_{\text{C-Net}}(\Theta^{t+1}(\phi^{t})) \right] \tag{22}$$

By Lipschitz smoothness of meta loss function, we have

$$\mathcal{L}_{\text{C-Net}}(\Theta^{t+2}(\phi^{t+1})) - \mathcal{L}_{\text{C-Net}}(\Theta^{t+1}(\phi^{t+1})) \leq \left\langle \nabla \mathcal{L}_{\text{C-Net}}(\Theta^{t+1}(\phi^{t+1})), \Theta^{t+2}(\phi^{t+1}) - \Theta^{t+1}(\phi^{t+1}) \right\rangle$$
$$+ \frac{L}{2} \left\| \Theta^{t+2}(\phi^{t+1}) - \Theta^{t+1}(\phi^{t+1}) \right\|_2^2 \tag{23}$$

According to the update rule, we have:

$$\Theta^{t+2}(\phi^{t+1}) - \Theta^{t+1}(\phi^{t+1}) = \left\langle \frac{\eta_1^{1t}}{n} \sum_{i=1}^n g_{\phi^t i}^{\alpha}[1] \cdot \nabla_{\theta_1^{t+1}} H\big(y_i^{(S_1)}, y_i\big) + g_{\phi^t i}^{\beta}[1] \cdot \tau^2 \cdot \nabla_{\theta_1^{t+1}} KL\big(y_i^{(S_1)}, y_i^{(T)}\big), \right.$$
$$\cdots,$$
$$\left. \frac{\eta_1^{kt}}{n} \sum_{i=1}^n g_{\phi^t i}^{\alpha}[k] \cdot \nabla_{\theta_k^{t+1}} H\big(y_i^{(S_k)}, y_i\big) + g_{\phi^t i}^{\beta}[k] \cdot \tau^2 \cdot \nabla_{\theta_k^{t+1}} KL\big(y_i^{(S_k)}, y_i^{(T)}\big) \right\rangle$$

This follows from Equation 4. Now, under the assumptions:

$$\left\| \nabla_{\theta_j^{t+1}} H\big(y_i^{(S_j)}, y_i\big) \right\| \leq \rho, \quad \left\| \nabla_{\theta_j^{t+1}} KL\big(y_i^{(S_j)}, y_i^{(T)}\big) \right\| \leq \rho,$$
$$\left\| \nabla H\big(f_{\theta_j}(x_i^v), y_i^v\big) \right\| \leq \rho, \quad g_{\phi}(\cdot) \leq 1, \quad \eta_1 = \max\{\eta_1^1, \dots, \eta_1^k\},$$

we obtain the following bound:

$$\left\| \mathcal{L}_{\text{C-Net}}\big(\Theta^{t+2}(\phi^{t+1})\big) - \mathcal{L}_{\text{C-Net}}\big(\Theta^{t+1}(\phi^{t+1})\big) \right\| \leq k\eta_1^t \rho^2 (1 + \tau^2) + \frac{kL(\eta_1^t(1+\tau^2))^2}{2}\rho^2$$
$$= k\eta_1^t \rho^2 (1 + \tau^2) \left( 1 + \frac{\eta_1^t L(1+\tau^2)}{2} \right) \tag{24}$$

By Lipschitz continuity of $\nabla \mathcal{L}_{\text{C-Net}}(\Theta^{t+1}(\phi))$ according to Lemma 1, we can obtain the following:

$$\mathcal{L}_{\text{C-Net}}(\Theta^{t+1}(\phi^{t+1})) - \mathcal{L}_{\text{C-Net}}(\Theta^{t+1}(\phi^{t})) \leq \left\langle \nabla \mathcal{L}_{\text{C-Net}}(\Theta^{t+1}(\phi^{t})), \phi^{t+1} - \phi^t \right\rangle + \frac{L}{2} \left\| \phi^{t+1} - \phi^t \right\|_2^2 \tag{25}$$

From Equation 21,

$$\mathcal{L}_{\text{C-Net}}(\Theta^{t+1}(\phi^{t+1})) - \mathcal{L}_{\text{C-Net}}(\Theta^{t+1}(\phi^{t})) \leq \left\langle \nabla \mathcal{L}_{\text{C-Net}}(\Theta^{t+1}(\phi^{t})), -\eta_2^t [\nabla_\phi \mathcal{L}_{\text{C-Net}}(\Theta^{t+1}(\phi^{t})) + \xi^{(t)}] \right\rangle$$
$$+ \frac{L(\eta_2^t)^2}{2} \left\| \nabla_\phi \mathcal{L}_{\text{C-Net}}(\Theta^{t+1}(\phi^{t})) + \xi^{(t)} \right\|_2^2 = -\left( \eta_2^t - \frac{L(\eta_2^t)^2}{2} \right) \left\| \nabla_\phi \mathcal{L}_{\text{C-Net}}(\Theta^{t+1}(\phi^{t})) \right\|$$
$$+ \frac{L(\eta_2^t)^2}{2} \|\xi^{(t)}\|_2^2 - (\eta_2^t - L(\eta_2^t)^2)\left\langle \nabla \mathcal{L}_{\text{C-Net}}(\Theta^{t+1}(\phi^{t})), \xi^{(t)} \right\rangle \tag{26}$$

Thus Equation 22 satifies,

$$\mathcal{L}_{\text{C-Net}}(\Theta^{t+2}(\phi^{t+1})) - \mathcal{L}_{\text{C-Net}}(\Theta^{t+1}(\phi^{t})) \leq k\eta_1^t \rho^2 (1 + \tau^2) \left( 1 + \frac{\eta_1^t L(1+\tau^2)}{2} \right)$$
$$- \left( \eta_2^t - \frac{L(\eta_2^t)^2}{2} \right) \left\| \nabla_\phi \mathcal{L}_{\text{C-Net}}(\Theta^{t+1}(\phi^{t})) \right\|_2^2$$
$$+ \frac{L(\eta_2^t)^2}{2} \|\xi^{(t)}\|_2^2 - (\eta_2^t - L(\eta_2^t)^2)\left\langle \nabla \mathcal{L}_{\text{C-Net}}(\Theta^{t+1}(\phi^{t})), \xi^{(t)} \right\rangle \tag{27}$$

Rearranging the terms, we obtain

$$
\left(\eta_2^t - \frac{L(\eta_2^t)^2}{2}\right)\left\|\nabla_\phi \mathcal{L}_{\text{C-Net}}(\Theta^{t+1}(\phi^t))\right\|_2^2 \leq \mathcal{L}_{\text{C-Net}}(\Theta^{t+2}(\phi^{t+1})) - \mathcal{L}_{\text{C-Net}}(\Theta^{t+1}(\phi^t))
$$
$$
+ k\eta_1^t \rho^2 (1+\tau^2)\left(1 + \frac{\eta_1^t L(1+\tau^2)}{2}\right)
$$
$$
+ \frac{L(\eta_2^t)^2}{2}\|\xi^{(t)}\|_2^2 - (\eta_2^t - L(\eta_2^t)^2)\langle \nabla\mathcal{L}_{\text{C-Net}}(\Theta^{t+1}(\phi^t)), \xi^{(t)}\rangle \tag{28}
$$

Summing up the above inequalities and rearranging the terms, we obtain

$$
\sum_{t=1}^{T}\left(\eta_2^t - \frac{L(\eta_2^t)^2}{2}\right)\left\|\nabla_\phi \mathcal{L}_{\text{C-Net}}(\Theta^{t+1}(\phi^t))\right\|_2^2 \leq \mathcal{L}_{\text{C-Net}}(\Theta^1(\phi^1)) - \mathcal{L}_{\text{C-Net}}(\Theta^{T+1}(\phi^T))
$$
$$
+ \sum_{t=1}^{T} k\eta_1^t \rho^2 (1+\tau^2)\left(1 + \frac{\eta_1^t L(1+\tau^2)}{2}\right) + \sum_{t=1}^{T}\frac{L(\eta_2^t)^2}{2}\|\xi^{(t)}\|_2^2
$$
$$
- \sum_{t=1}^{T}(\eta_2^t - L(\eta_2^t)^2)\left\langle \nabla\mathcal{L}_{\text{C-Net}}(\Theta^{t+1}(\phi^t)), \xi^{(t)}\right\rangle
$$
$$
\leq \mathcal{L}_{\text{C-Net}}(\Theta^1(\phi^1)) + \sum_{t=1}^{T} k\eta_1^t \rho^2 (1+\tau^2)\left(1 + \frac{\eta_1^t L(1+\tau^2)}{2}\right) + \sum_{t=1}^{T}\frac{L(\eta_2^t)^2}{2}\|\xi^{(t)}\|_2^2
$$
$$
- \sum_{t=1}^{T}(\eta_2^t - L(\eta_2^t)^2)\langle \nabla\mathcal{L}_{\text{C-Net}}(\Theta^{t+1}(\phi^t)), \xi^{(t)}\rangle \tag{29}
$$

Taking expectations with respect to $\xi^{(N)}$ on both sides, we obtain:

$$
\sum_{t=1}^{T}\left(\eta_2^t - \frac{L(\eta_2^t)^2}{2}\right)\mathbb{E}_{\xi^{(N)}}\left\|\nabla_\phi \mathcal{L}_{\text{C-Net}}(\Theta^{t+1}(\phi^t))\right\|_2^2 \leq \mathcal{L}_{\text{C-Net}}(\Theta^1(\phi^1))
$$
$$
+ \sum_{t=1}^{T} k\eta_1^t \rho^2 (1+\tau^2)\left(1 + \frac{\eta_1^t L(1+\tau^2)}{2}\right) + \sum_{t=1}^{T}\frac{L(\eta_2^t)^2}{2}\|\xi^{(t)}\|_2^2 \tag{30}
$$

since $\mathbb{E}_{\xi^{(N)}}\langle \nabla\mathcal{L}_{\text{C-Net}}(\Theta^{t+1}(\phi^t)), \xi^{(t)}\rangle = 0$ and $\mathbb{E}[\|\xi^{(t)}\|_2^2] \leq \sigma^2$, where $\sigma^2$ is the variance of $\xi^{(t)}$. Furthermore, we can deduce that

$$
\min_t \mathbb{E}\left\|\nabla_\phi \mathcal{L}_{\text{C-Net}}(\Theta^{t+1}(\phi^t))\right\|_2^2 \leq \frac{\sum_{t=1}^{T}\left(\eta_2^t - \frac{L(\eta_2^t)^2}{2}\right)\mathbb{E}_{\xi^{(N)}}\left\|\nabla_\phi \mathcal{L}_{\text{C-Net}}(\Theta^{t+1}(\phi^t))\right\|_2^2}{\sum_{t=1}^{T}\left(\eta_2^t - \frac{L(\eta_2^t)^2}{2}\right)}
$$
$$
\leq \frac{1}{\sum_{t=1}^{T}(2\eta_2^t - L(\eta_2^t)^2)}\left[2\mathcal{L}_{\text{C-Net}}(\Theta^1(\phi^1)) + \sum_{t=1}^{T}k\eta_1^t\rho^2(1+\tau^2)(2+\eta_1^t L(1+\tau^2))\right.
$$
$$
\left. + L\sigma^2\sum_{t=1}^{T}(\eta_2^t)^2\right]
$$
$$
\leq \frac{1}{\sum_{t=1}^{T}\eta_2^t}\left[2\mathcal{L}_{\text{C-Net}}(\Theta^1(\phi^1)) + \sum_{t=1}^{T}k\eta_1^t\rho^2(1+\tau^2)(2+\eta_1^t L(1+\tau^2)) + L\sigma^2\sum_{t=1}^{T}(\eta_2^t)^2\right]
$$
$$
\leq \frac{1}{T\eta_2^t}\left[2\mathcal{L}_{\text{C-Net}}(\Theta^1(\phi^1)) + Tk\eta_1^1\rho^2(1+\tau^2)(2+\eta_1^1 L(1+\tau^2)) + L\sigma^2\sum_{t=1}^{T}(\eta_2^t)^2\right]
$$

$$\leq \frac{2\mathcal{L}_{\text{C-Net}}(\Theta^1(\phi^1))}{T\eta_2^t} + \frac{k\eta_1^1\rho^2(1+\tau^2)(2+L(1+\tau^2))}{\eta_2^t} + L\sigma^2\eta_2^t$$

$$= \frac{\mathcal{L}_{\text{C-Net}}(\Theta^1(\phi^1))}{T}\max\{L, \frac{\sigma\sqrt{T}}{c}\}$$

$$+ \min\{1, \frac{a}{T}\}\max\{L, \frac{\sigma\sqrt{T}}{c}\}k\rho^2(1+\tau^2)(2+L(1+\tau^2)) + L\sigma^2\min\{\frac{1}{L}, \frac{c}{\sigma\sqrt{T}}\}$$

$$\leq \frac{\mathcal{L}_{\text{C-Net}}(\Theta^1(\phi^1))}{c\sqrt{T}} + \frac{a\sigma k\rho^2(1+\tau^2)(2+L(1+\tau^2))}{c\sqrt{T}} + \frac{L\sigma c}{\sqrt{T}} = \mathcal{O}(\frac{1}{\sqrt{T}}). \tag{31}$$

The third inequality holds for $\sum_{t=1}^T (2\eta_2^t - L(\eta_2^t)^2) \leq \sum_{t=1}^T \eta_2^t$. Therefore, we can conclude that our algorithm can always achieve $\min_{0\leq t\leq T}\mathbb{E}[\|\nabla\mathcal{L}_{\text{C-Net}}(\Theta^1)\|_2^2] \leq \mathcal{O}(\frac{1}{\sqrt{T}})$ in $T$ steps, and this finishes our proof of Theorem 1.

$\blacksquare$

**Lemma 2** *(Lemma A.5 in (Mairal, 2013)) Let $(a_n)_{n\leq 1}, (b_n)_{n\leq 1}$ be two non-negative real sequences such that the series $\sum_{i=1}^\infty a_n$ diverges, the series $\sum_{i=1}^\infty a_n b_n$ converges, and there exists $K > 0$ such that $|b_{n+1} - b_n| \leq Ka_n$. Then the sequences $(b_n)_{n\leq 1}$ converges to 0.*

**Theorem 2** *Suppose the loss function $\mathcal{L}$ is Lipschitz smooth with constant $L$, and C-Net$(\cdot)$ is differential with a $\delta$-bounded gradient and twice differential with its Hessian bounded by $\mathcal{B}$, and the loss function $\mathcal{L}$ have $\rho$-bounded gradients with respect to training/meta data. Let the learning rates $\eta_1^{1t}, \cdots, \eta_1^{kt}$ satisfies $\eta_1^{jt} = \min\{1, \frac{a}{T}\}\forall j \in \{1, \cdots, k\}$, for some $a > 0$, such that $\frac{a}{T} < 1$, and $\eta_2^t, 1 \leq t \leq N$ is a monotone descent sequence, $\eta_2^t = \min\{\frac{1}{L}, \frac{c}{\sigma\sqrt{T}}\}$ for some $c > 0$, such that $\frac{\sigma\sqrt{T}}{c} \geq L$ and $\sum_{t=1}^\infty \eta_2^t \leq \infty, \sum_{t=1}^\infty (\eta_2^t)^2 \leq \infty$. Then*

$$\lim_{t\to\infty}\mathbb{E}[\|\nabla\mathcal{L}_{S_j}(\theta_j^t, \phi^t)\|_2^2] = 0. \tag{32}$$

**Proof**

From Equation 4 we have,

$$\mathcal{L}_{S_j}(\theta_j^t, \phi^t) = \frac{1}{n}\sum_{i=1}^n g_{\phi^t i}^\alpha[j] * H\big(y_i^{(S_j)}, y_i\big) + g_{\phi^t i}^\beta[j] * \tau^2 * KL\big(y_i^{(S_j)}, y_i^{(T)}\big) \quad \forall j \in \{1, \cdots, k\} \tag{33}$$

It is easy to conclude that $\eta jt_1$ satisfy $\sum_{t=0}^\infty \eta_1^{jt} = \infty, \sum_{t=0}^\infty (\eta_1^{jt})^2 < \infty$. From Equation 4

$$\theta_j^{t+1} = \theta_j^t - \frac{\eta_1^j}{n}\sum_{i=1}^n g_{\phi^t i}^\alpha[j] * \nabla_{\theta_j^t} H\big(y_i^{(S_j)}, y_i\big) + g_{\phi^t i}^\beta[j] * \tau^2 * \nabla_{\theta_j^t} KL\big(y_i^{(S_j)}, y_i^{(T)}\big) \quad \forall j \in \{1, \cdots, k\} \tag{34}$$

Since the mini-batch $\Psi_t$ is drawn uniformly at random, we can rewrite the update equation as:

$$\theta_j^{t+1} = \theta_j^t - \frac{\eta_1^j}{n}\sum_{i=1}^n \Big[ g_{\phi^t i}^\alpha[j] * \nabla_{\theta_j^t} H\big(y_i^{(S_j)}, y_i\big)$$

$$+ g_{\phi^t i}^\beta[j] * \tau^2 * \nabla_{\theta_j^t} KL\big(y_i^{(S_j)}, y_i^{(T)}\big) + \Psi_t \Big] \quad \forall j \in \{1, \cdots, k\}$$

$$= \theta_j^t - \eta_1^{tj}[\nabla\mathcal{L}_{S_j}(\theta_j^t, \phi^t) + \Psi_t] \tag{35}$$

where $\psi^{(t)} = \nabla\mathcal{L}_{S_j}(\theta_j^t, \phi^t)|_{\Psi_t} - \mathcal{L}_{S_j}(\theta_j^t, \phi^t)$. Note that $\psi^{(t)}$ is i.i.d. random variable with finite variance, since $\Psi_t$ are drawn i.i.d. with a finite number of samples. Furthermore, $\mathbb{E}[\psi^{(t)}] = 0$, since samples are drawn uniformly at random, and $\mathbb{E}[\|\psi^{(t)}\|_2^2] \leq \sigma^2$.

The objective function $\mathcal{L}_{S_j}(\theta_j^t, \phi^t)$ can be easily checked to be Lipschitz-smooth with constant $L$, and have $\rho$-bounded gradients with respect to training data. Observe that,

$$\mathcal{L}_{S_j}(\theta_j^{t+1}, \phi^{t+1}) - \mathcal{L}_{S_j}(\theta_j^t, \phi^t) = \left\{ \mathcal{L}_{S_j}(\theta_j^{t+1}, \phi^{t+1}) - \mathcal{L}_{S_j}(\theta_j^{t+1}, \phi^t) \right\} + \left\{ \mathcal{L}_{S_j}(\theta_j^{t+1}, \phi^t) - \mathcal{L}_{S_j}(\theta_j^t, \phi^t) \right\} \quad (36)$$

For the first term,

$$\mathcal{L}_{S_j}(\theta_j^{t+1}, \phi^{t+1}) - \mathcal{L}_{S_j}(\theta_j^{t+1}, \phi^t) = \frac{1}{n} \sum_{i=1}^n (g_{\phi^{t+1}i}^\alpha[j] - g_{\phi^t i}^\alpha[j]) * H(y_i^{(S_j)}, y_i)$$

$$+ (g_{\phi^{t+1}i}^\beta[j] - g_{\phi^t i}^\beta[j]) * \tau^2 * KL(y_i^{(S_j)}, y_i^{(T)})$$

$$\leq \frac{1}{n} \sum_{i=1}^n \left\{ \langle \nabla_\phi g_{\phi^t i}^\alpha[j], \phi^{t+1} - \phi^t \rangle + \frac{\delta(\eta_2^t)^2}{2} \|\phi^{t+1} - \phi^t\|_2^2 \right\} * H(y_i^{(S_j)}, y_i)$$

$$+ (g_{\phi^{t+1}i}^\beta[j] - g_{\phi^t i}^\beta[j]) * \tau^2 * KL(y_i^{(S_j)}, y_i^{(T)})$$

$$= \frac{1}{n} \sum_{i=1}^n \left\{ \langle \nabla_\phi g_{\phi^t i}^\alpha[j], -\eta_2^t[\nabla_\phi \mathcal{L}_{\text{C-Net}}(\Theta^{t+1}(\phi^t)) + \xi^{(t)}] \rangle \right.$$

$$\left. + \frac{\delta(\eta_2^t)^2}{2} \|\nabla_\phi \mathcal{L}_{\text{C-Net}}(\Theta^{t+1}(\phi^t)) + \xi^{(t)}\|_2^2 \right\} * H(y_i^{(S_j)}, y_i) + (g_{\phi^{t+1}i}^\beta[j] - g_{\phi^t i}^\beta[j]) * \tau^2 * KL(y_i^{(S_j)}, y_i^{(T)})$$

$$= \frac{1}{n} \sum_{i=1}^n \left\{ \langle \nabla_\phi g_{\phi^t i}^\alpha[j], -\eta_2^t[\nabla_\phi \mathcal{L}_{\text{C-Net}}(\Theta^{t+1}(\phi^t)) + \xi^{(t)}] \rangle + \frac{\delta(\eta_2^t)^2}{2} \left( \|\nabla_\phi \mathcal{L}_{\text{C-Net}}(\Theta^{t+1}(\phi^t))\|_2^2 + \|\xi^{(t)}\|_2^2 \right. \right.$$

$$\left. \left. + 2\langle \nabla_\phi \mathcal{L}_{\text{C-Net}}(\Theta^{t+1}(\phi^t)), \xi^{(t)} \rangle \right) \right\} * H(y_i^{(S_j)}, y_i) + (g_{\phi^{t+1}i}^\beta[j] - g_{\phi^t i}^\beta[j]) * \tau^2 * KL(y_i^{(S_j)}, y_i^{(T)})$$

$$\leq \frac{1}{n} \sum_{i=1}^n \left\{ \langle \nabla_\phi g_{\phi^t i}^\alpha[j], -\eta_2^t[\nabla_\phi \mathcal{L}_{\text{C-Net}}(\Theta^{t+1}(\phi^t)) + \xi^{(t)}] \rangle + \frac{\delta(\eta_2^t)^2}{2} \left( \|\nabla_\phi \mathcal{L}_{\text{C-Net}}(\Theta^{t+1}(\phi^t))\|_2^2 + \|\xi^{(t)}\|_2^2 \right. \right.$$

$$\left. \left. + 2\langle \nabla_\phi \mathcal{L}_{\text{C-Net}}(\Theta^{t+1}(\phi^t)), \xi^{(t)} \rangle \right) \right\} * H(y_i^{(S_j)}, y_i) + \frac{1}{n} \sum_{i=1}^n \left\{ \langle \nabla_\phi g_{\phi^t i}^\beta[j], -\eta_2^t[\nabla_\phi \mathcal{L}_{\text{C-Net}}(\Theta^{t+1}(\phi^t)) + \xi^{(t)}] \rangle \right.$$

$$\left. + \frac{\delta(\eta_2^t)^2}{2} \left( \|\nabla_\phi \mathcal{L}_{\text{C-Net}}(\Theta^{t+1}(\phi^t))\|_2^2 + \|\xi^{(t)}\|_2^2 + 2\langle \nabla_\phi \mathcal{L}_{\text{C-Net}}(\Theta^{t+1}(\phi^t)), \xi^{(t)} \rangle \right) \right\} * \tau^2 * KL(y_i^{(S_j)}, y_i^{(T)}) \quad (37)$$

For the second term from Equation 36,

$$\mathcal{L}_{S_j}(\theta_j^{t+1}, \phi^t) - \mathcal{L}_{S_j}(\theta_j^t, \phi^t) \leq \langle \nabla_{\theta_j} \mathcal{L}_{S_j}(\theta_j^t, \phi^t), \theta_j^{t+1} - \theta_j^t \rangle + \frac{L(\eta_1^j)^2}{2} \|\theta_j^{t+1} - \theta_j^t\|_2^2$$

$$= \langle \nabla_{\theta_j} \mathcal{L}_{S_j}(\theta_j^t, \phi^t), -\eta_1^{tj}[\nabla \mathcal{L}_{S_j}(\theta_j^t, \phi^t) + \Psi_t] \rangle + \frac{L(\eta_1^j)^2}{2} \|\nabla \mathcal{L}_{S_j}(\theta_j^t, \phi^t) + \Psi_t\|_2^2$$

$$= -\left( \eta_1^{tj} - \frac{L(\eta_1^j)^2}{2} \right) \|\nabla \mathcal{L}_{S_j}(\theta_j^t, \phi^t)\|_2^2 + \frac{L(\eta_1^j)^2}{2} \|\psi^{(t)}\|_2^2$$

$$- (\eta_1^{tj} - L(\eta_1^j)^2) \langle \nabla \mathcal{L}_{S_j}(\theta_j^t, \phi^t), \psi^{(t)} \rangle \quad (38)$$

Combining Equations 37 and 38 we can rewrite Equation 36 as,

$$\mathcal{L}_{S_j}(\theta_j^{t+1}, \phi^{t+1}) - \mathcal{L}_{S_j}(\theta_j^t, \phi^t) \leq \frac{1}{n} \sum_{i=1}^n \left\{ \langle \nabla_\phi g_{\phi^t i}^\alpha[j], -\eta_2^t[\nabla_\phi \mathcal{L}_{\text{C-Net}}(\Theta^{t+1}(\phi^t)) + \xi^{(t)}] \rangle \right.$$

$$\left. + \frac{\delta(\eta_2^t)^2}{2} \left( \|\nabla_\phi \mathcal{L}_{\text{C-Net}}(\Theta^{t+1}(\phi^t))\|_2^2 + \|\xi^{(t)}\|_2^2 + 2\langle \nabla_\phi \mathcal{L}_{\text{C-Net}}(\Theta^{t+1}(\phi^t)), \xi^{(t)} \rangle \right) * H(y_i^{(S_j)}, y_i) \right.$$

$$+ \langle \nabla_\phi g_{\phi^t i}^\beta[j], -\eta_2^t[\nabla_\phi \mathcal{L}_{\text{C-Net}}(\Theta^{t+1}(\phi^t)) + \xi^{(t)}] \rangle$$

$$+ \frac{\delta(\eta_2^t)^2}{2}\left(\|\nabla_\phi \mathcal{L}_{\text{C-Net}}(\Theta^{t+1}(\phi^t))\|_2^2 + \|\xi^{(t)}\|_2^2 + 2\langle \nabla_\phi \mathcal{L}_{\text{C-Net}}(\Theta^{t+1}(\phi^t)), \xi^{(t)}\rangle\right) * \tau^2 * KL\left(y_i^{(S_j)}, y_i^{(T)}\right)\Big\}$$

$$- \left(\eta_1^{tj} - \frac{L(\eta_1^j)^2}{2}\right)\|\nabla \mathcal{L}_{S_j}(\theta_j^t, \phi^t)\|_2^2 + \frac{L(\eta_1^j)^2}{2}\|\psi^{(t)}\|_2^2 - (\eta_1^{tj} - L(\eta_1^j)^2)\langle \nabla \mathcal{L}_{S_j}(\theta_j^t, \phi^t), \psi^{(t)}\rangle \tag{39}$$

Taking expectation on both sides and since $\mathbb{E}[\xi^{(t)}] = 0, \mathbb{E}[\psi^{(t)}] = 0$, we have

$$\mathbb{E}[\mathcal{L}_{S_j}(\theta_j^{t+1}, \phi^{t+1})] - \mathbb{E}[\mathcal{L}_{S_j}(\theta_j^t, \phi^t)] \leq \mathbb{E}\left[\frac{1}{n}\sum_{i=1}^n\left\{\langle \nabla_\phi g_{\phi_i^t}^\alpha[j], -\eta_2^t \nabla_\phi \mathcal{L}_{\text{C-Net}}(\Theta^{t+1}(\phi^t))\rangle\right.\right.$$

$$\left.\left. + \frac{\delta(\eta_2^t)^2}{2}\left(\|\nabla_\phi \mathcal{L}_{\text{C-Net}}(\Theta^{t+1}(\phi^t))\|_2^2 + \|\xi^{(t)}\|_2^2\right)\right\} * H\left(y_i^{(S_j)}, y_i\right)\right]$$

$$+ \mathbb{E}\left[\frac{1}{n}\sum_{i=1}^n\left\{\langle \nabla_\phi g_{\phi_i^t}^\beta[j], -\eta_2^t \nabla_\phi \mathcal{L}_{\text{C-Net}}(\Theta^{t+1}(\phi^t))\rangle\right.\right.$$

$$\left.\left. + \frac{\delta(\eta_2^t)^2}{2}\left(\|\nabla_\phi \mathcal{L}_{\text{C-Net}}(\Theta^{t+1}(\phi^t))\|_2^2 + \|\xi^{(t)}\|_2^2 + \right) * \tau^2 * KL\left(y_i^{(S_j)}, y_i^{(T)}\right)\right\}\right]$$

$$- \eta_1^j \mathbb{E}[\|\nabla \mathcal{L}_{S_j}(\theta_j^t, \phi^t)\|_2^2] + \frac{L(\eta_1^j)^2}{2}\left(\mathbb{E}[\|\nabla \mathcal{L}_{S_j}(\theta_j^t, \phi^t)\|_2^2] + \mathbb{E}[\|\psi^{(t)}\|_2^2]\right) \tag{40}$$

Summing up the above inequalities over $t = 1, ..., \infty$ in both sides, we obtain,

$$\sum_{t=1}^\infty \eta_1^{tj}\mathbb{E}[\|\nabla \mathcal{L}_{S_j}(\theta_j^t, \phi^t)\|_2^2] + \sum_{t=1}^\infty \eta_2^t \mathbb{E}\left[\frac{1}{n}\sum_{i=1}^n \|H\left(y_i^{(S_j)}, y_i\right)\|\|\nabla_\phi g_{\phi_i^t}^\alpha[j]\|.\|\nabla_\phi \mathcal{L}_{\text{C-Net}}(\Theta^{t+1}(\phi^t))\|\right]$$

$$+ \sum_{t=1}^\infty \eta_2^t \tau^2 \mathbb{E}\left[\frac{1}{n}\sum_{i=1}^n \|KL\left(y_i^{(S_j)}, y_i^{(T)}\right)\|\|\nabla_\phi g_{\phi_i^t}^\beta[j]\|.\|\nabla_\phi \mathcal{L}_{\text{C-Net}}(\Theta^{t+1}(\phi^t))\|\right]$$

$$\leq \sum_{t=1}^\infty \frac{L(\eta_1^j)^2}{2}\left(\mathbb{E}[\|\nabla \mathcal{L}_{S_j}(\theta_j^t, \phi^t)\|_2^2] + \mathbb{E}[\|\psi^{(t)}\|_2^2]\right) + \mathbb{E}[\mathcal{L}_{S_j}(\theta_j^1, \phi^1)] - \lim_{T\to\infty}\mathbb{E}[\mathcal{L}_{S_j}(\theta_j^t, \phi^t)]$$

$$+ \sum_{t=1}^\infty \frac{\delta(\eta_2^t)^2}{2}\left\{\frac{1}{n}\sum_{i=1}^n\left(\mathbb{E}\|\nabla_\phi \mathcal{L}_{\text{C-Net}}(\Theta^{t+1}(\phi^t))\|_2^2 + \mathbb{E}\|\xi^{(t)}\|_2^2\right)\left(\|H\left(y_i^{(S_j)}, y_i\right)\| + \tau^2\|KL\left(y_i^{(S_j)}, y_i^{(T)}\right)\|\right)\right\}$$

$$\leq \sum_{t=1}^\infty \frac{L(\eta_1^j)^2}{2}[\rho^2 + \sigma^2] + \mathbb{E}[\mathcal{L}_{S_j}(\theta_j^1, \phi^1)] + \sum_{t=1}^\infty \frac{\delta(\eta_2^t)^2}{2}\left\{2\tau^2 * M(\rho^2 + \sigma^2)\right\} \leq \infty \tag{41}$$

The last inequality holds since $\sum_{t=0}^\infty (\eta_1^j)^2 < \infty, \sum_{t=0}^\infty (\eta_2^t)^2 < \infty, 1 + \tau^2 \approx \tau^2$ and $\frac{1}{n}\sum_{i=1}^n \|H\left(y_i^{(S_j)}, y_i\right)\| \leq M$ and $\frac{1}{n}\sum_{i=1}^n \|KL\left(y_i^{(S_j)}, y_i^{(T)}\right)\| \leq M$ for limited number of samples' loss is bounded. Thus, we have

$$\sum_{t=1}^\infty \eta_1^{tj}\mathbb{E}[\|\nabla \mathcal{L}_{S_j}(\theta_j^t, \phi^t)\|_2^2] + \sum_{t=1}^\infty \eta_2^t \mathbb{E}\left[\frac{1}{n}\sum_{i=1}^n \|H\left(y_i^{(S_j)}, y_i\right)\|\|\nabla_\phi g_{\phi_i^t}^\alpha[j]\|.\|\nabla_\phi \mathcal{L}_{\text{C-Net}}(\Theta^{t+1}(\phi^t))\|\right]$$

$$+ \sum_{t=1}^\infty \eta_2^t \tau^2 \mathbb{E}\left[\frac{1}{n}\sum_{i=1}^n \|KL\left(y_i^{(S_j)}, y_i^{(T)}\right)\|\|\nabla_\phi g_{\phi_i^t}^\beta[j]\|.\|\nabla_\phi \mathcal{L}_{\text{C-Net}}(\Theta^{t+1}(\phi^t))\|\right] \leq \infty \tag{42}$$

Since

$$\sum_{t=1}^\infty \eta_2^t \mathbb{E}\left[\frac{1}{n}\sum_{i=1}^n \|H\left(y_i^{(S_j)}, y_i\right)\|\|\nabla_\phi g_{\phi_i^t}^\alpha[j]\|.\|\nabla_\phi \mathcal{L}_{\text{C-Net}}(\Theta^{t+1}(\phi^t))\|\right] \leq M\delta\rho \sum_{t=1}^\infty \eta_2^t \leq \infty \tag{43}$$

and

$$\sum_{t=1}^\infty \eta_2^t \tau^2 \mathbb{E}\left[\frac{1}{n}\sum_{i=1}^n \|KL\left(y_i^{(S_j)}, y_i^{(T)}\right)\|\|\nabla_\phi g_{\phi_i^t}^\beta[j]\|.\|\nabla_\phi \mathcal{L}_{\text{C-Net}}(\Theta^{t+1}(\phi^t))\|\right] \leq \tau^2 M\delta\rho \sum_{t=1}^\infty \eta_2^t \leq \infty \tag{44}$$

This implies that $\sum_{t=1}^{\infty} \eta_1^{tj} \mathbb{E}[\|\nabla\mathcal{L}_{S_j}(\theta_j^t, \phi^t)\|_2^2] < \infty$.

By Lemma 2, to substantiate $\lim_{t\to\infty} \mathbb{E}[\|\nabla\mathcal{L}_{S_j}(\theta_j^t, \phi^t)\|_2^2] = 0$ since $\sum_{t=0}^{\infty} \eta_1^{tj} = \infty$, therefore we only need to show,

$$\left| \mathbb{E}[\|\nabla\mathcal{L}_{S_j}(\theta_j^{t+1}, \phi^{t+1})\|_2^2 - \mathbb{E}[\|\nabla\mathcal{L}_{S_j}(\theta_j^t, \phi^t)\|_2^2]] \right| \leq C\eta_1^j \tag{45}$$

for some constant $C$. Based on the inequality:

$$|(\|a\| + \|b\|)(\|a\| - \|b\|)| \leq \|a + b\|\|a - b\|, \tag{46}$$

we then have:

$$
\begin{aligned}
\left| \mathbb{E}[\|\nabla\mathcal{L}_{S_j}(\theta_j^{t+1}, \phi^{t+1})\|_2^2 - \mathbb{E}[\|\nabla\mathcal{L}_{S_j}(\theta_j^t, \phi^t)\|_2^2]] \right| &= \left| (\mathbb{E}[\|\nabla\mathcal{L}_{S_j}(\theta_j^{t+1}, \phi^{t+1})\|_2 \right. \\
&\quad \left. - \mathbb{E}[\|\nabla\mathcal{L}_{S_j}(\theta_j^t, \phi^t)\|_2]])(\mathbb{E}[\|\nabla\mathcal{L}_{S_j}(\theta_j^{t+1}, \phi^{t+1})\|_2] + \mathbb{E}[\|\nabla\mathcal{L}_{S_j}(\theta_j^t, \phi^t)\|_2]) \right| \\
&\leq \mathbb{E}\left[ \left| \|\nabla\mathcal{L}_{S_j}(\theta_j^{t+1}, \phi^{t+1})\|_2 - \|\nabla\mathcal{L}_{S_j}(\theta_j^t, \phi^t)\|_2 \right| \left| \|\nabla\mathcal{L}_{S_j}(\theta_j^{t+1}, \phi^{t+1})\|_2 + \|\nabla\mathcal{L}_{S_j}(\theta_j^t, \phi^t)\|_2 \right| \right] \\
&\leq \mathbb{E}\left[ \|\nabla\mathcal{L}_{S_j}(\theta_j^{t+1}, \phi^{t+1}) - \nabla\mathcal{L}_{S_j}(\theta_j^t, \phi^t)\|_2 \|\nabla\mathcal{L}_{S_j}(\theta_j^{t+1}, \phi^{t+1}) + \nabla\mathcal{L}_{S_j}(\theta_j^t, \phi^t)\|_2 \right] \\
&\leq \mathbb{E}\left[ (\|\nabla\mathcal{L}_{S_j}(\theta_j^{t+1}, \phi^{t+1})\|_2 - \|\nabla\mathcal{L}_{S_j}(\theta_j^t, \phi^t)\|_2)(\|\nabla\mathcal{L}_{S_j}(\theta_j^{t+1}, \phi^{t+1})\| + \|\nabla\mathcal{L}_{S_j}(\theta_j^t, \phi^t)\|_2) \right] \\
&\leq 2L\rho\mathbb{E}\left[ \|(\theta_j^{t+1}, \phi^{t+1}) - (\theta_j^t, \phi^t)\|_2 \right] \\
&\leq 2L\rho\eta_2^t\eta_1^{tj}\mathbb{E}\left[ \|(\nabla\mathcal{L}_{S_j}(\theta_j^t, \phi^t) + \Psi_t, \nabla_\phi\mathcal{L}_{\text{C-Net}}(\Theta^{t+1}(\phi^t)) + \xi^{(t)})\|_2 \right] \\
&\leq 2L\rho\eta_2^t\eta_1^{tj}\mathbb{E}\left[ \sqrt{\|(\nabla\mathcal{L}_{S_j}(\theta_j^t, \phi^t) + \Psi_t\|_2^2} + \sqrt{\|\nabla_\phi\mathcal{L}_{\text{C-Net}}(\Theta^{t+1}(\phi^t)) + \xi^{(t)}\|_2^2} \right] \\
&\leq 2L\rho\eta_2^t\eta_1^{tj}\sqrt{\mathbb{E}[\|(\nabla\mathcal{L}_{S_j}(\theta_j^t, \phi^t) + \Psi_t\|_2^2] + \mathbb{E}[\|\nabla_\phi\mathcal{L}_{\text{C-Net}}(\Theta^{t+1}(\phi^t)) + \xi^{(t)}\|_2^2]} \\
&\leq 2L\rho\eta_2^t\eta_1^{tj}\sqrt{\mathbb{E}[\|(\nabla\mathcal{L}_{S_j}(\theta_j^t, \phi^t)\|_2^2 + \mathbb{E}[\Psi_t\|_2^2] + \mathbb{E}[\|\nabla_\phi\mathcal{L}_{\text{C-Net}}(\Theta^{t+1}(\phi^t))\|_2^2] + \mathbb{E}[\xi^{(t)}\|_2^2]} \\
&\leq 2L\rho\eta_2^t\eta_1^{tj}\sqrt{2\sigma^2 + 2\rho^2} \tag{47}
\end{aligned}
$$

According to the above inequality, we can conclude that our algorithm can achieve

$$\lim_{t\to\infty} \mathbb{E}[\|\nabla\mathcal{L}_{S_j}(\theta_j^t, \phi^t)\|_2^2] = 0 \tag{48}$$

The proof is completed.

∎

# B  Training Details

## B.1  Dataset Details

We conduct experiments using the following real-world datasets to showcase the effectiveness of our approach,

**CIFAR-100** (Krizhevsky, 2009). The dataset consists of a total of 60K examples, distributed across 100 distinct classes. Each example in this dataset comprises images with a resolution of $32 \times 32 \times 3$. Specifically, the training set encompasses 50,000 examples, while the remaining 10K serve as the testing set. In our experimental setup, approximately 5K examples are allocated for use as a validation set for our C-Net. For the other baseline models, these validation examples are utilized for hyper-parameter tuning.

**ImageNet-1K** (Russakovsky et al., 2015). The dataset comprises over 1.2 million training images and 50,000 validation images, spanning 1,000 object categories. Each image is high-resolution and varies in size, but is typically resized to $224 \times 224 \times 3$ during preprocessing. In our experiments, we follow the standard ImageNet-1K split. A subset of 50K images from the training set is held out as a validation set for tuning the parameters of the coordinator network (C-Net). For the remaining baselines, this same subset is used exclusively for hyperparameter tuning.

**Tiny ImageNet** (Le & Yang, 2015). The dataset is derived from the extensive ImageNet-1K dataset (Russakovsky et al., 2015). This dataset encompasses a total of 100K images, which have been downsampled to a $64 \times 64$ resolution, representing a subset of 200 classes mirroring those in the ImageNet-1K dataset, with each class containing precisely 500 images. As part of our experimental protocol, we have set aside an independent validation set comprising 10K examples, which is utilized for both our proposed method and the baseline models.

**Clothing-1M** (Xiao et al., 2015). The dataset consists of 10,00,000 noisy label training examples spread across 14 class labels of clothing items taken from various online shopping websites. Each example consists of an image with resolution $224 \times 224 \times 3$. The validation and test set consists of 14,313 and 10,526 examples respectively. The dataset also contains 47,560 clean examples, but we do not use that portion for our experiments. It is a dataset with noisy labels, since the data is collected from several online shopping websites and include many mis-labelled samples.

**Instance CIFAR-100** (Xia et al., 2020) The dataset contains manually corrupted version (with an error rate $\tau$) of the CIFAR-100 (Krizhevsky, 2009) dataset. This is done by first constructing a transition matrix $\mathcal{T}(\mathbf{x})$, where $\mathcal{T}_{ij}(\mathbf{x})$ is the probability that the corrupted label is $j$, given the true label is $i$; while keeping the overall error rate as $\tau$. Then, the labels are flipped according to the defined transition probabilties. In our experiments, the training uses 50K examples with noisy labels, corrupted by the above method, while the remaining 10K examples, which serve as testing set have no label noise. Approximately 5K noisy examples are allocated for use as a validation set for our C-Net. The overall label flip rate $\tau$, used in our experiments was 0.2.

**iWildCam-2020** (Beery et al., 2020) The dataset is set of collection of images collected using the heat trap and motion activated cameras for the better understanding of wildlife and biodiversity. There is so wide diversity in lighting, camera angle, backdrop, vegetation, color, and relative animal frequencies between different camera traps, and also the number of animals of a particular species varies a lot, hence the data about different spices also vary similarly. In our experimentation we trained the models using approx 100k training, 12k validation and 12k test images across 216 classes.

**Tiny ImageNet-C** (Hendrycks & Dietterich, 2019). The dataset is a downsampled version of the ImageNet-C dataset, which consists of 15 diverse corruption types applied to the ImageNet-1K (Russakovsky et al., 2015) validation dataset. These corruptions have been drawn from 4 main categories - noise, blur, weather, and digital. Each corruption has 5 severity levels, 1 being the least severe and 5 being the most severe. In our experiments, the models trained on Tiny ImageNet (Le & Yang, 2015) dataset, with train-validation split of 90K and 10K examples respectively, are tested on Tiny ImageNet-C for testing robustness gains.

**MS COCO** (Lin et al., 2014) The dataset contains over 330K images, with more than 200K labeled and 80K annotated with object segmentation masks, spanning 80 object categories. Each image typically includes multiple objects in complex scenes with rich contextual information. In our experimental setup, we use the standard training-validation split, selecting a subset of 5K images from the training set as a validation set for tuning C-Net. For all other baselines, this validation set is used solely for hyperparameter optimization.

For all the image classification datasets, we use the data augmentation methods, using the torchvision's transforms module. We use RandomCrop, RandomResizedCrop, RandomSizedCrop, RandomHorizontalFlip, Normalize, and ColorJitter for our purpose. Augmentation methods were applied on both datasets, over all experiments, baselines, overall models.

| Architecture | Filters | Basic block Repeats | CIFAR-100 | | Tiny ImageNet | |
|---|---|---|---|---|---|---|
| | | | MACs | Params | MACs | Params |
| ResNet10-xxs | [8, 8, 16, 16] | [1, 1, 1, 1] | 2 M | 13 K | 8 M | 15 K |
| ResNet10-xs | [8, 16, 16, 32] | [1, 1, 1, 1] | 3 M | 28 K | 12 M | 31 K |
| ResNet10-s | [8, 16, 32, 64] | [1, 1, 1, 1] | 4 M | 84 K | 16 M | 90 K |
| ResNet10-m | [16, 32, 64, 128] | [1, 1, 1, 1] | 16 M | 320 K | 64 M | 333 K |
| ResNet10-l | [32, 64, 128, 256] | [1, 1, 1, 1] | 64 M | 1.25 M | 255 M | 1.28 M |
| ResNet10 | [64, 128, 256, 512] | [1, 1, 1, 1] | 253 M | 4.92 M | 1013 M | 5 M |
| ResNet18 | [64, 128, 256, 512] | [2, 2, 2, 2] | 555 M | 11.22 M | 2221 M | 11.27 M |
| ResNet34 | [64, 128, 256, 512] | [3, 4, 6, 3] | 1159 M | 21.32 M | 4637 M | 21.38 M |

Table 13: Comparision of newly introduced smaller ResNet with the standard ResNet models

## B.2 Model Details

**Image Classification Models** In Section 4.1, we introduce several smaller ResNet models, specifically ResNet10-xxxs, ResNet10-xxs, ResNet10-xs, ResNet10-s, ResNet10-m, and ResNet10-l. These models are adopted from Kag et al. (2023), and in Table 13, we compare their architectural details with well-established ResNet models like ResNet10, ResNet18, and ResNet34. Similar to the standard ResNet models, these newer, more compact versions also employ the traditional 'BasicBlock' as their fundamental building block. The architectural structure consists of a convolutional block, four stages of residual blocks, an adaptive average pooling layer, a convolutional block, and a classifier layer. The sole variation among the various capacity variants within this experimental setup is the number of filters in each stage and the residual block. Additionally, Table 13 provides information about the number of parameters and multiply-addition (MAC) operations for each model for the datasets CIFAR-100 and Tiny-ImageNet.

**Object Detection Models** For the object detection experiments in Section 4.2, we employ Faster R-CNN (Ren et al., 2015) as the base detector, with a high-capacity ResNet50-FPN as the teacher, and ResNet18-FPN and ResNet34-FPN as the student models. These are equipped with Feature Pyramid Networks (FPNs) (Lin et al., 2017) that enhance multi-scale feature learning. The C-Net used in this setting is a ResNet34 model, modified to output weighting parameters in place of classification logits. All models are initialized from ImageNet-pretrained weights and are fine-tuned on the MS COCO dataset following standard detection pipelines. Unlike the plain ResNet architectures used for image classification, Faster R-CNN incorporates a *Region Proposal Network* (RPN) and a detection head on top of a ResNet-FPN backbone. The ResNet-FPN backbone outputs multi-scale feature maps, enabling detection of objects at various sizes. These features are processed by the RPN to generate region proposals, which are then classified and refined by the ROI head.

## B.3 Hyper-Parameters

For all the KD baselines listed in Section 4.1.2 and MC-DISTIL, we use temperature $\tau = 2$, employ the SGD optimizer to train the student models and ADAM optimizer (Kingma & Ba, 2014) to train C-NET. We use a batch size of 400 for all datasets. We train the student models for 300 epochs on CIFAR-100, Instance CIFAR-100, Clothing-1M, Tiny ImageNet and Tiny ImageNet-C datasets; and for 100 epochs on iWildCam dataset, ImageNet and MS COCO while updating C-NET every 20 epoch (*i.e.*, L = 20). We use cosine annealing (Loshchilov & Hutter, 2017) as the learning rate schedule for training the student models. We warm start each student model by first training it using the cross-entropy loss without using the teacher model for all KD baselines and MC-DISTIL. For all datasets, we perform a grid search on 0.1, 0.05 for the learning rate, on $1e-5, 5e-5, 1e-4$ for the weight decay, and 0.65, 0.75, 0.85, 0.95 for the momentum. For the C-NET training we use a learning rate of $1e-3$ and set weight decay to $1e-4$.

### B.4 Baselines

We compare MC-DISTIL with the following SOTA Knowledge distillation baselines:

**Teacher Assistant Knowledge Distillation (TAKD)** (Mirzadeh et al., 2020) This approach introduces a multi-step distillation process that leverages intermediate-level teachers to facilitate the efficient transfer of knowledge from a large pre-trained teacher network to a more compact student model. To realize this, we employ a teacher network identical to our own and enlist fellow student models with higher learning capacities to serve as intermediate models in this knowledge distillation process.

**Densely Guided Knowledge Distillation (DGKD)** (Son et al., 2021) Much like TAKD, this approach employs several intermediate models; however, it distinguishes itself by training the final student model through a single distillation step. In addition to the teacher KL divergence loss, the training objective for the final student model incorporates the KL divergence loss obtained from the pre-trained intermediate models.

**Robust Model Compression (RMC)** (Du et al., 2023). It uses multiple students with various levels of sparsity and interprets the variance in their predictions for each instance as a measure of task complexity. Subsequently, it refines the teacher predictions based on this complexity metric, resulting in a more robust knowledge distillation process. In our experiments, we employ students with diverse learning capacities as a substitute for models with different levels of sparsity, achieving similar benefits.

**Deep Mutual Information (DML)** (Zhang et al., 2018) It is a collaborative learning paradigm that leverages the interactions among multiple neural networks to enhance their collective performance. In DML, a group of neural networks, often referred to as student models, collaboratively learns by exchanging information with each other during training. This is achieved by using the predictions or feature representations of one model to guide the learning process of others. The collaborative nature of DML allows the models to exploit the diverse knowledge learned by their peers, fostering a mutual learning dynamic.

**SHAdow KnowlEdge transfer framework (SHAKE)** (Li & Jin, 2022) It is a novel knowledge distillation framework designed to enhance the transfer of knowledge from a teacher model to a student model in deep learning. Introduced to address the challenges associated with large capacity gaps between teacher and student models, SHAKE employs a pseudo-teacher concept. This pseudo-teacher consists of a set of additional learnable layers attached to the backbone of the teacher model. What sets SHAKE apart is its ability to dynamically adapt to the student model during training.

In experiments where we train with noisy datasets, we compare against additional two bilevel optimisation-based reweighting schemes:

**Learning to Reweight (L2R)** (Ren et al., 2018) It is a machine learning paradigm that focuses on training models to dynamically assign weights to different instances in a dataset during the learning process. The fundamental idea behind L2R is to adaptively adjust the significance of individual data points, emphasizing more informative instances and de-emphasizing noisy or less relevant ones. This approach is particularly valuable in scenarios where the training data may be imbalanced or subject to noise, helping models prioritize the most crucial samples for better generalization.

**Meta-Weight-Net (MWN)** (Shu et al., 2019) Like L2R, MWN also learns weights assigned to individual samples based on their specific characteristics or importance. However, this explicit mapping of sample weights is learned through a meta-learning framework. Meta-Weight-Net aims to address challenges related to imbalanced or noisy datasets by explicitly learning a mapping function that adapts sample weights during the training process. By doing so, the model can assign higher importance to informative or challenging samples and reduce the impact of less relevant ones.

### B.5 Compute Resources

We our experiments on a mixture of GPUS *viz.* A100s and RTX 2080 as our experiments don't need any sophisticated modern GPUs. MC-DISTIL memory requirements are slightly higher that traditional KD as we train student cohorts simultaneously. The additional memory requirements depend the number of students used and size of cohorts.

| | CE | KD | TAKD | DGKD | RMC | DML | SHAKE | MetaDistil |
|---|---|---|---|---|---|---|---|---|
| KD | 1.53e-05 | | | | | | | |
| TAKD | 1.53e-05 | 0.0005 | | | | | | |
| DGKD | 1.53e-05 | 1.53e-05 | 0.0006 | | | | | |
| RMC | 4.78e-05 | 0.99 | 0.9992 | 1 | | | | |
| DML | 1.53e-05 | 0.978 | 1 | 1 | 0.1156 | | | |
| SHAKE | 4.78e-05 | 0.115 | 0.64714 | 0.9855 | 0.0222 | 0.0416 | | |
| MetaDistil | 1.53e-05 | 1.53e-05 | 0.0005 | 0.5896 | 4.58e-05 | 1.53e-05 | 0.001 | |
| MC-Distil | 1.53e-05 | 1.53e-05 | 1.53e-05 | 7.63e-05 | 1.53e-05 | 1.53e-05 | 0.00015 | 1.53e-05 |

Table 14: Pairwise significance p-values using Wilcoxon signed rank test

### B.6 Timing Calculation Methodology

The values presented in Table 12 are based on profiling a full epoch of training and are intended to offer a fair estimate of the computational budget required to train an entire cohort of student models. The following principles were applied:

A standard heuristic was used to estimate computational cost: a model's forward pass contributes $1\times$ its theoretical FLOPs, while a full training step—comprising both forward and backward passes—is estimated to consume $3\times$ the forward FLOPs, following Zhou et al. (2021).

The profiling was performed on the full student cohort consisting of ResNet10-xxs, ResNet10-xs, ResNet10-s, and ResNet10-m, with ResNet34 used as the teacher. All reported FLOPs are normalized on a per-epoch basis. Auxiliary components were configured as follows: SHAKE uses a ResNet18 proxy teacher per student; MetaDistil uses a ResNet32 meta-network (similar to AMAL); and MC-Distil uses a ResNet32-based C-NET (C-Net). RMC relies on the cohort itself for difficulty estimation.

For sequential distillation methods such as TAKD (Teacher $\rightarrow$ TA$_1$ $\rightarrow$ TA$_2$ ...) and DGKD (Teacher + TAs $\rightarrow$ TA), total per-epoch time and TFLOPs are calculated as the cumulative sum across each training stage. This approach captures the actual total training effort.

For methods that train each student independently—such as SHAKE, MetaDistil, and RMC—the total TFLOPs are computed as the sum of each student-specific run. However, the reported time per epoch reflects the slowest student model (ResNet10-m), representing an optimistic scenario under ideal hardware parallelism.

In contrast, MC-Distil trains all students simultaneously in a single run, reusing shared components such as the C-NET, thereby significantly reducing both wall-clock time and aggregate computational cost.

## C Additional Experiments

We have released the code at the following URL: `https://github.com/AtharvaTambat/MC-Distil`

### C.1 Standard deviation and statistical significance results:

In Table 14, we show the p-values of one-tailed Wilcoxon signed-rank test Wilcoxon (1992) performed on every single possible pair of knowledge distillation strategies to determine whether there is a significant statistical difference between the strategies in each pair, across all datasets. Our null hypothesis is that there is no difference between the knowledge distillation strategies pair. From the results, it is evident that MC-DISTIL significantly outperforms other baselines at $p < 0.05$.

Table 15 shows the standard deviation results over three training runs on CIFAR-100 and Tiny ImageNet datasets. The results show that the MC-DISTIL has the least standard deviation compared two of the most competitive baselines viz. DGKD and MetaDistil. We note that DGKD has large standard deviations owing to not having a principled way of introducing student cohort models.

| CIFAR100 Test Accuracies | | | | | | | |
|---|---|---|---|---|---|---|---|
| Teacher | | Student | CE | KD | DGKD | MetaDistil | MC-Distil |
| ResNet10 | 75.18 | ResNet10-xxs | $32.41 \pm 0.517$ | $35.23 \pm 1.132$ | $34.57 \pm 0.453$ | $35.47 \pm 0.737$ | $36.23 \pm 0.247$ |
| | | ResNet10-xs | $42.42 \pm 0.724$ | $45.62 \pm 0.722$ | $44.89 \pm 1.775$ | $46.86 \pm 0.836$ | $47.74 \pm 0.244$ |
| | | ResNet10-s | $52.70 \pm 0.384$ | $55.64 \pm 0.106$ | $56.25 \pm 0.545$ | $57.25 \pm 0.535$ | $58.42 \pm 0.275$ |
| | | ResNet10-m | $64.19 \pm 0.142$ | $67.05 \pm 0.191$ | $67.27 \pm 0.577$ | $68.75 \pm 0.185$ | $69.45 \pm 0.427$ |
| ResNet18 | 76.99 | ResNet10-xxs | $32.41 \pm 0.517$ | $35.19 \pm 1.415$ | $34.14 \pm 1.259$ | $35.26 \pm 0.745$ | $36.03 \pm 0.442$ |
| | | ResNet10-xs | $42.42 \pm 0.724$ | $45.49 \pm 0.412$ | $46.18 \pm 1.055$ | $46.23 \pm 0.420$ | $47.24 \pm 0.459$ |
| | | ResNet10-s | $52.70 \pm 0.384$ | $55.49 \pm 0.315$ | $56.07 \pm 0.578$ | $57.26 \pm 0.131$ | $57.63 \pm 0.176$ |
| | | ResNet10-m | $64.19 \pm 0.142$ | $66.42 \pm 0.095$ | $66.73 \pm 0.548$ | $68.02 \pm 0.490$ | $68.52 \pm 0.237$ |
| Tiny-ImageNet Test Accuracies | | | | | | | |
| ResNet10 | 44.04 | ResNet10-xxs | $13.68 \pm 0.460$ | $14.40 \pm 0.526$ | $14.20 \pm 0.475$ | $14.47 \pm 0.597$ | $15.33 \pm 0.521$ |
| | | ResNet10-xs | $18.78 \pm 0.231$ | $20.20 \pm 0.624$ | $20.90 \pm 0.351$ | $20.85 \pm 0.692$ | $22.07 \pm 0.285$ |
| | | ResNet10-s | $24.86 \pm 0.362$ | $27.83 \pm 0.779$ | $27.59 \pm 0.605$ | $28.07 \pm 0.890$ | $29.62 \pm 0.263$ |
| | | ResNet10-m | $33.93 \pm 0.493$ | $35.86 \pm 0.332$ | $35.82 \pm 0.398$ | $37.64 \pm 1.452$ | $38.46 \pm 0.068$ |
| ResNet18 | 47.94 | ResNet10-xxs | $13.68 \pm 0.460$ | $14.52 \pm 0.368$ | $14.42 \pm 0.481$ | $14.68 \pm 0.595$ | $15.26 \pm 0.105$ |
| | | ResNet10-xs | $18.78 \pm 0.231$ | $20.28 \pm 0.500$ | $20.58 \pm 0.931$ | $20.95 \pm 1.038$ | $21.36 \pm 0.360$ |
| | | ResNet10-s | $24.96 \pm 0.362$ | $27.26 \pm 0.874$ | $27.82 \pm 0.378$ | $28.45 \pm 0.983$ | $30.36 \pm 0.006$ |
| | | ResNet10-m | $33.93 \pm 0.493$ | $35.9 \pm 0.724$ | $35.96 \pm 0.612$ | $37.82 \pm 1.562$ | $39.11 \pm 0.295$ |

Table 15: Standard deviation results for CIFAR-100 and Tiny ImageNet datasets for 3 runs. We compare MC-Distil with two most competitive baselines.

## C.2 Robustness of MC-Distil trained models

For the models trained with Resnet18 serving as the teacher model on the Tiny ImageNet dataset in Table 1, we present their performance on Tiny ImageNet-C (Hendrycks & Dietterich, 2019) (see Appendix B.1) in Table 16 incorporating all baseline models and diverse corruption levels. We assess the resilience of models whose perfromance was reported in Table 1 when confronted with data distribution mismatches. Notably, across various corruption levels and teacher-student combinations, MC-Distil demonstrates robust performance, even though it was not explicitly tailored for handling distributional shifts between training and test datasets. Specifically, at lower corruption levels, models trained with MC-Distil exhibit a performance advantage of 1-2% compared to other baselines. Even at higher corruption levels, MC-Distil maintains competitiveness with the baselines.

## D Limitations & Social Impact

We have presented a novel *meta-collaborative distillation*—where students of different capacities are codistilled from a single teacher. While the initial results wide range of student & teacher architectures and model sizes are encouraging, more work is needed to understand under what conditions such adaptive mixing optimizations for metacollaborative distillation can provide gains. Our experiments are proof-of-concept on widely used benchmarking datasets; future work will explore whether similar gains can be realized in real-world applications as well. Our proposal involves a bi-level optimization objective (for which we provide a cheap approximate meta-learning preocedure)—the tradeoff of increased training time vs increased accuracy may depend on the specific application context.

Our work is a fairly general-purpose optimization procedure for improving knowledge transfer from a teacher model to a very small student model and therefore tremendously improving the inference time. This could yield incremental improvements to a broad range of ML applications, and as a result benefit those applications. We do not anticipate any inherent risk of negative social impact from our work, over and above the risk inherent in the ML application itself.

| Tiny ImageNet-C Test Accuracies | | | | | | | | |
|---|---|---|---|---|---|---|---|---|
| **Student** | **CE** | **KD** | **MCD** | **DGKD** | **DML** | **SKAKE** | **MetaDistil** | **MC-Distil** |
| Corruption level 1 Accuracies | | | | | | | | |
| RN10-xxs | 11.25 | 11.38 | 10.61 | 10.77 | 09.99 | 11.47 | 11.02 | **12.05** |
| RN10-xs | 14.68 | 15.82 | 14.67 | 15.43 | 15.51 | 16.56 | 16.02 | **16.73** |
| RN10-s | 20.88 | 21.97 | 21.94 | 23.18 | 20.29 | 22.58 | 23.97 | **24.88** |
| RN10-m | 26.91 | 28.56 | 27.08 | 29.56 | 28.09 | 30.40 | 30.43 | **31.00** |
| Corruption level 2 Accuracies | | | | | | | | |
| RN10-xxs | 10.65 | 10.79 | 09.54 | 10.42 | 09.44 | 10.88 | 10.52 | **11.30** |
| RN10-xs | 13.75 | 14.60 | 13.72 | 14.60 | 14.56 | 15.42 | 14.78 | **15.52** |
| RN10-s | 19.28 | 20.64 | 20.46 | 21.26 | 18.74 | 20.74 | 21.93 | **23.04** |
| RN10-m | 24.76 | 25.87 | 24.36 | 26.97 | 26.20 | 27.94 | 27.39 | **28.67** |
| Corruption level 3 Accuracies | | | | | | | | |
| RN10-xxs | 08.66 | 08.78 | 07.54 | 08.93 | 08.46 | 09.29 | 08.58 | **09.42** |
| RN10-xs | 10.94 | 11.54 | 11.08 | 11.82 | 11.96 | 12.19 | 11.78 | **12.58** |
| RN10-s | 15.68 | 16.85 | 16.71 | 16.95 | 14.76 | 16.95 | 17.65 | **18.79** |
| RN10-m | 19.70 | 21.07 | 18.79 | 22.20 | 21.45 | 22.23 | 20.83 | **22.63** |
| Corruption level 4 Accuracies | | | | | | | | |
| RN10-xxs | 04.92 | 04.66 | 04.38 | 05.52 | 05.05 | 05.57 | 04.60 | **05.87** |
| RN10-xs | 06.77 | 06.96 | 05.88 | **07.54** | 06.89 | 06.57 | 07.35 | 07.39 |
| RN10-s | 09.41 | 09.92 | 10.21 | 09.42 | 08.35 | 10.29 | 09.77 | **10.80** |
| RN10-m | 11.12 | 12.28 | 10.74 | 12.74 | **12.87** | 12.44 | 11.54 | 12.03 |
| Corruption level 5 Accuracies | | | | | | | | |
| RN10-xxs | 03.58 | 03.15 | 03.44 | 03.98 | 03.66 | 04.29 | 03.33 | **04.40** |
| RN10-xs | 05.14 | 05.18 | 04.34 | **06.23** | 05.15 | 05.05 | 05.47 | 05.53 |
| RN10-s | 06.94 | 07.24 | 07.33 | 07.07 | 06.19 | **07.93** | 07.15 | 07.38 |
| RN10-m | 08.16 | 09.09 | 07.86 | 09.73 | 09.46 | **09.48** | 08.44 | 08.89 |

Table 16: Comprehensive table presenting an assessment of model (trained with Tiny ImageNet in Table 1) robustness by conducting inference on the Tiny ImageNet-C dataset (Hendrycks & Dietterich, 2019) across increasing degrees of image corruption (from level 1 to 5), alongside results for all baseline models. The columns indicate the method employed for training the model on the Tiny ImageNet dataset, utilizing ResNet18 as the teacher model. The highest accuracies are highlighted in bold.

