# OpenReview forum: "Unified Wisdom: Harnessing Collaborative Learning to Improve Efficacy of Knowledge Distillation"
_TMLR — Accepted by TMLR_

### Review · Reviewer_qANo · 2025-06-03

**Summary Of Contributions:**

This paper introduces Meta-Collaborative Distillation (MC-DISTIL), a knowledge distillation framework designed to address distillation when a significant capacity gap exists between teacher and student models. The core idea is to facilitate collaborative learning among a cohort of student models with varying capacities, all learning from a single teacher. This collaboration is orchestrated by a Coordinator Network (C-NET), which functions as a meta-learner. C-NET modulates the training loss for each student on an instance-by-instance basis by reweighting the standard cross-entropy loss and the teacher-matching KL divergence loss. The C-NET learns from each student's performance and training instance characteristics, enabling students of different capacities to improve together. The authors demonstrate that MC-DISTIL achieves significant accuracy gains on datasets like CIFAR100 and TinyImageNet across diverse student and teacher architectures, outperforming existing state-of-the-art methods.

**Audience:**

Yes

**Claims And Evidence:**

Yes

**Requested Changes:**

1. Computational Cost Analysis:

    Provide a more explicit discussion or table comparing the computational cost (e.g., training time, FLOPs relative to training the smallest student model with cross-entropy, or relative to standard KD) of MC-DISTIL against key baseline methods. This would help readers better understand the trade-offs involved.

2. ImageNet Experimentation:

    The authors are encouraged to provide results for MC-DISTIL on ImageNet with more traditional teacher-student pairings, such as ResNet101 to ResNet50 or ResNet101 to ResNet34. This would offer a clearer picture of its benefits in commonly studied large-scale distillation scenarios.

3. Terminology Clarification:

    Revise the statement on page 4, section 3.1 regarding labels. Instead of "Typically, the labels y are cardinal," consider "Typically, in classification, labels y are categorical, often numerically encoded". Ordinal labels could also be mentioned as a specific type of categorical label.

4. Address Typos:

**Strengths And Weaknesses:**

**Strengths**
* Novel Approach: The concept of meta-collaborative distillation using a C-NET to modulate learning for a cohort of diverse students is innovative. It moves beyond traditional single-student distillation or multi-teacher setups.
* Addressing Capacity Gaps: The method specifically targets and shows strong performance in scenarios with large teacher-student capacity gaps, a known limitation in KD.
* Improved Performance: MC-DISTIL demonstrates consistent and significant average accuracy gains over state-of-the-art baselines on benchmark datasets (e.g., 2.5% on CIFAR100, 2% on TinyImageNet). All students in the cohort, regardless of size, show near consistent  improvement.
* Implicit Instance Hardness Learning: The C-NET implicitly learns to differentiate instance hardness and tailors learning for students accordingly (e.g., weaker students focus on easier examples).

**Weaknesses**
- Computational Cost: Training multiple students simultaneously along with an additional C-NET introduces increased computational overhead in terms of training time and memory compared to standard KD. While the paper mentions an "efficient alternating update algorithm" and updating C-NET less frequently, a more explicit comparison of computational costs against baselines is missing.

- ImageNet Results Interpretation: While the ImageNet results in Table 3 demonstrate MC-DISTIL's effectiveness in extreme teacher-student gap settings (e.g., ResNet101 to ResNet14/20/32/56), the choice of student models makes the improvements less directly comparable to more traditional distillation scenarios with full models such as ResNet101 teacher and ResNet50, ResNet34 students. The reported student accuracies are also relatively low for ImageNet, though this is acknowledged as a consequence of the large gap and even CE performing poorly.

**Typos and Minor Corrections**
1. Location: Page 4, Section 3.1 ("Standard Knowledge Distillation"), Sentence 2

    Original Text: "Typically, the labels y are cardinal, and practitioners have found that such labels are often inadequate in capturing nuances in the data."

    Suggested Correction: The term "cardinal" here might be less precise. For classification, labels are typically "categorical." "Ordinal" would apply if there's an inherent order to the categories. "Cardinal" refers to the count of elements in a set (e.g., the number of unique classes). A more accurate phrasing might be: "Typically, in classification, labels y are categorical..."

2. Location: Page 5, Section 3.2.1, First paragraph
    Original Text: "...as the C-NETiS a centralized mechanism..."

    Typo: C-NETiS

     Suggested Fix: C-NET is
3. Location: Page 5, Section 3.2.1, First paragraph

    Original Text: "...represent the loss mixing parameters for the k students on the input xi​.Therefore, the ..."

    Typo: input $x_i$.Therefore

    Suggested Fix: input $x_i$. Therefore

4. Location: Page 9, Section 4.5, Second paragraph

    Original Text: "...choosing a favorable ground truth is also important along with focusing on the "learnable" points."

    Formatting Suggestion: "learnable" -> ``learnable'' (i.e., using LaTeX-style double backticks and double apostrophes for opening and closing quotes).

5. Location: Page 10, Section 4.7 ("KD in Instance Dependent Label-Noise"), First paragraph

    Original Text: "...instance CIFAR-100 (Xia et al., 2020a) and Clothing-1M Xiao et al. (2015b) where the labels inherit noise..."

    Citation Style Suggestion: For "Xiao et al. (2015b)", consider using \citet (or the appropriate command for the paper's citation package) if the goal is to have the author names as part of the sentence text, e.g., "Clothing-1M by Xiao et al. (2015b)".

---

> ### Author Response · Authors · 2025-06-19
>
> We express our heartfelt gratitude to the reviewer for their dedicated time and thorough evaluation of our paper. The valuable insights they offered are highly appreciated, and we are grateful for the constructive feedback. We are devoted to addressing the reviewer's concerns and integrating their suggestions into the final version to elevate the overall quality of our paper. The following section presents comprehensive responses to the reviewer's comments, queries, and recommendations to enhance the paper's overall rating.
>
> ``computational costs``
>
>  Compared to the baselines such as TAKD, DGKD and DML, which use multiple student models during training, MC-Distil additionally introduces a C-Net. For a student pool of ResNet10-xxs, ResNet10-xs, ResNet10-s, and ResNet10-m, ResNet34 as the teacher model, and ResNet10-l as the C-Net we incur just 5% of additional FLOPS. Further, since the reduction in C-Net size doesn’t affect the MC-Distil performance drastically (as shown in appendix C.2) the C-Net size could be further optimized. Both C-Net and student cohorts are presented only during the training phase. Our method doesn’t require any ensemble framework or multistudent framework at the time of inference; therefore, our method has no additional cost compared to traditional KD during inference.
>
> ``ImageNet results``
>
> Training a ResNet-50 model is computationally expensive, especially on the ImageNet dataset. Due to limited time and compute resources, we adopt a standard distillation setup on the ImageNet dataset, where ResNet-34 serves as the teacher and ResNet-18 as the student. Given our focus on a multi-student distillation setting, we follow the approach introduced in the DKGD paper [1], which incorporates ResNet-26 as an intermediate student model to facilitate more effective knowledge transfer.
>
>  Teacher   | Teacher Acc | Student   | CE    | KD    | TAKD  | DGKD  | MC-Distil |
> |-----------|-------------|-----------|-------|-------|-------|-------|----------|
> |           |             | ResNet18  | 69.75 | 70.66 | 71.37 | 71.73 | **71.95** |
> | ResNet34  | 73.3        | ResNet26  | 66.61 | 67.38 |       |       | **69.31** |
>
> We observe that the performance gains from MC-Distil for ResNet-18 are smaller compared to those for ResNet-26. This is likely because ResNet-18 is a well-established model on ImageNet, with extensively optimized training recipes, leaving less room for additional improvements. In contrast, ResNet-26 is less commonly used and lacks such fine-tuned setups, allowing MC-Distil to have a greater impact. This also suggests that MC-Distil can be particularly valuable in reducing the need for extensive hyperparameter tuning, especially for under-optimized or custom architectures.
>
> To further assess MC-Distil's robustness under resource constraints, we conduct additional experiments where a ResNet-101 teacher is used to distill into ResNet-50, ResNet-34, and ResNet-18 students, but with a key modification: all student models are trained using 64×64 input resolution, as opposed to the standard 224×224, mimicking low-resource or low-resolution deployment settings. The teacher model, however, remains trained on standard 224×224 inputs.
>
> | Teacher    | Teacher Acc | Student   | CE    | KD    | MCDistil |
> |------------|-------------|-----------|-------|-------|----------|
> |            |             | ResNet50  | 62.95 | 65.59 | **66.31** |
> | ResNet101  | 81.68       | ResNet34  | 62.89 | 64.89 | **65.12** |
> |            |             | ResNet18  | 58.16 | 58.36 | **59.20** |
>
> Here again MC-Distil consistently improves performance across all students compared to both baseline cross-entropy (CE) training and standard knowledge distillation (KD). These results demonstrate that MC-Distil remains effective even when student models operate on low-resolution inputs, transferring knowledge from a high-capacity, high-resolution teacher.
>
> [1] Son, W., Na, J., Choi, J., & Hwang, W. (2021). Densely guided knowledge distillation using multiple teacher assistants. In Proceedings of the IEEE/CVF International Conference on Computer Vision (pp. 9395-9404).

---

> ### Author Response · Authors · 2025-06-22
> **Typographical, Stylistic, and Citation Corrections**
>
> We thank the reviewer for the careful reading and helpful suggestions. We have addressed all the noted issues to improve the clarity and consistency of the manuscript. Specifically:
>
> - **Page 4, Section 3.1**: We replaced the term *"cardinal"* with *"categorical"* to more accurately describe classification labels, as per the reviewer’s suggestion.
>
> - **Page 5, Section 3.2.1**: The typo *"C-NETiS"* has been corrected to *"C-NET is"*, and the spacing error in *"input xi.Therefore"* has been fixed to read *"input xi. Therefore"*.
>
> - **Page 9, Section 4.5**: We reformatted the word *“learnable”* using LaTeX-style quotation marks (``learnable'') to maintain typographic consistency.
>
> - **Page 10, Section 4.7**: The citation for *"Xiao et al. (2015b)"* has been updated to ensure consistency with the surrounding citation style.
>
> We appreciate the reviewer’s attention to these details, which helped improve the overall quality and presentation of the manuscript.

---

> ### Author Response · Authors · 2025-06-22
> **Computational Cost and Time-Based Analysis**
>
> We sincerely thank the reviewer for suggesting a computational and time-based analysis. This was very helpful, and we believe the added comparison strengthens our work by highlighting the efficiency of our method.
>
> As recommended, we profiled several distillation techniques. The table below summarizes the total computational cost (in TFLOPs) and training time required to train all student models for one epoch.
>
> | Method            | GFLOPs per Batch | Total TFLOPs per Epoch | Per-Epoch Training Time (s) |
> |-------------------|------------------|-------------------------|-----------------------------|
> | **TAKD [1]**      | 1.49             | 0.58                    | 20.06                       |
> | **DGKD [2]**      | 4.77             | 1.86                    | 24.63                       |
> | **RMC [3]**       | 4.74             | 1.85                    | 12.49                       |
> | **SHAKE [4]**     | 18.63            | 7.26                    | 14.39                       |
> | **MetaDistill [5]** | 10.29          | 4.01                    | 9.10                        |
> | **MC-Distil**     | 2.47             | 0.97                    | 11.90                       |
>
> ### Key Observations
>
> - **Sequential Bottleneck:** Methods such as **TAKD** [1] and **DGKD** [2] require the sequential training of progressively smaller models. Consequently, their total training time is the sum of the times for each stage, making them significantly slower at producing the full cohort of students compared to our simultaneous approach.
>
> - **Auxiliary Model Overhead:** Methods like **SHAKE** [4] and **MetaDistill** [5], which achieve per-epoch training times comparable to **MC-Distil**, rely on auxiliary models that must be trained separately for each student. For example, SHAKE [4] requires a distinct proxy teacher for each student, while MetaDistill [5] uses a separate meta-network for loss re-weighting. To train the same number of students as our model, these approaches must be executed independently for each student, making them more computationally expensive in terms of total required resources and FLOPs.
>
> - **Accuracy vs. Efficiency:** While methods like **RMC** [3] are computationally lighter than some of the more complex meta-learning strategies, our proposed **MC-Distil** significantly outperforms it in terms of final model accuracy, as demonstrated in the table below. This highlights our method’s ability to achieve a superior balance of efficiency and performance.
>
> ### Accuracy Comparison with ResNet34 Teacher
>
> To further contextualize these findings, the following table presents a direct accuracy comparison between **RMC** [3] and **MC-Distil** on CIFAR-100 and Tiny-ImageNet, using ResNet34 as the teacher. This setup corresponds to the computational analysis described above (Further details about setup are in the next section).
>
> | Dataset         | Student Model | RMC      | **MC-Distil (Ours)** |
> |------------------|----------------|----------|-----------------------|
> | **CIFAR-100**    | ResNet10-xxs   | 34.46%   | **36.10%**            |
> |                  | ResNet10-xs    | 42.78%   | **47.12%**            |
> |                  | ResNet10-s     | 53.58%   | **57.67%**            |
> |                  | ResNet10-m     | 65.58%   | **68.24%**            |
> | **Tiny-ImageNet**| ResNet10-xxs   | 13.78%   | **15.24%**            |
> |                  | ResNet10-xs    | 19.28%   | **21.82%**            |
> |                  | ResNet10-s     | 25.99%   | **29.59%**            |
> |                  | ResNet10-m     | 33.58%   | **38.34%**            |
>
> This analysis underscores that our method not only offers significant computational advantages but also consistently achieves higher accuracy across all student models and datasets in this setting.

---

> > ### Author Response · Authors · 2025-06-22
> > **Computational Cost and Time-Based Analysis**
> >
> > ### Calculation Methodology
> >
> > The values presented in the table are based on profiling the run for one epoch, and are designed to provide a fair comparison of the total computational cost required to train a full cohort of student models. The following principles were applied:
> >
> > 1. A standard heuristic was used to estimate the computational cost of network operations. A model's forward pass contributes `1x` its theoretical FLOPs. A full training step, comprising both a forward and a backward pass for gradient computation, is estimated to contribute `3x` its forward pass FLOPs [6].
> >
> > 2. The experimental setup profiles the total time and workload required to train the entire student cohort, which consists of `ResNet10_m`, `ResNet10_s`, `ResNet10_xs`, and `ResNet10_xxs`. The primary teacher model is a ResNet34 architecture. All reported metrics—both time and FLOPs—are normalized to a per-epoch basis. The auxiliary models are specified as follows: the proxy teacher in **SHAKE** [4] is a `ResNet18`; the meta-model in **MetaDistill** [5] (similar to AMAL [5]) and the C-Net in **MC-Distil** are both `ResNet32`; and the cohort of student models is used for difficulty estimation in **RMC** [3].
> >
> > 3. For sequential distillation methods such as **TAKD** [1] (Teacher → TA₁ → TA₂ ...) and **DGKD** [2] (Teacher + TAs → next TA), the total per-epoch time and FLOPs are calculated as the **sum of the costs of each required stage**. This cumulative value represents the total workload to complete one epoch of training for every model in the sequential chain.
> >
> > 4. For methods that require independent training sessions for each student, such as **SHAKE** [4], **MetaDistill** [5], and **RMC** [3], the following approach was taken to represent total workload versus ideal parallel execution:
> >    - The **Total FLOPs** values are calculated as the **sum of the computational costs for all required independent runs** needed to train the entire student cohort. This provides a true measure of the total computational budget.
> >    - The **Total Time** reported is the profiled wall-clock time for only the **most computationally expensive run** (i.e., the one involving `ResNet10_m`). This metric represents the best-case parallel execution time, assuming sufficient hardware to run all student trainings concurrently.
> >
> > ### References
> >
> > [1] Mirzadeh, S. I. et al. (2020). *Improved knowledge distillation via teacher assistant.* AAAI 2020.
> > [2] Son, W. et al. (2021). *Densely guided knowledge distillation using multiple teacher assistants.* ICCV 2021.
> > [3] Du, M. et al. (2023). *Robustness challenges in model distillation and pruning for natural language understanding.* EACL 2023.
> > [4] Li, L. & Jin, Z. (2022). *Shadow knowledge distillation: Bridging offline and online knowledge transfer.* NeurIPS 2022.
> > [5] Sivasubramanian, D. et al. (2023). *Adaptive mixing of auxiliary losses in supervised learning.* AAAI 2023.
> >
> > [6] Zhou, X. et al. (2021). Efficient neural network training via forward and backward propagation sparsification. NeurIPS 2021 (arXiv:2111.05685).

---

### Review · Reviewer_b4Eo · 2025-06-04

**Summary Of Contributions:**

This paper introduces a collaborative knowledge distillation approach where multiple student models of varying capacities learn simultaneously from a single teacher, coordinated by an auxiliary "coordinator network" (C-Net). The key innovation is that C-Net dynamically reweights loss components for each student on a per-instance basis, allowing weaker students to focus on easier examples while stronger students tackle harder cases. Notably, the framework demonstrates that smaller students can actually help improve larger students through this collaborative process.

**Audience:**

Yes

**Claims And Evidence:**

No

**Requested Changes:**

Please consider the weaknesses. I would recommend that the authors consult the cited works [1,2,3,4] and consider reporting on the various benchmarks and evaluation settings used in those works. I would additionally request detailed ablations on the C-Net and potential variants to understand its effectiveness.

**Strengths And Weaknesses:**

Strengths:

1. The method is well-motivated and innovative particularly the coordinator network concept.
2. The paper is well written and easy to follow.

Weaknesses:

I have a two major concerns:

1. There is no clear ablation on the impact of C-Net. For example, what if one was to use independent learnable loss weights for each student individually, rather than generating them through the proposed C-Net? Or use weights proportional to the capacity of the student? There needs to be clarity on the impact of the C-Net module. As an additional example, the authors mention that a naive solution of using a n × k matrix would be prohibitive as training samples increase. Would it be possible to try out this method on CIFAR100 only, where the training set is quite small? Just as a comparison of how good such a strategy can be regardless of the tradeoff.

2. The experimental analysis is a bit limited. Results are limited to only classification, mainly with TinyImageNet and CIFAR100. Similarly, the networks are mainly limited to various ResNet variants. It would be interesting to see the performance beyond classification (such as object detection on MSCOCO in various other works [1,2,3,4]) and different networks including the cases where teacher and student are different architectures [2]. Without detailed experimentation, the effectiveness of the method is in question.

[1] Chen, P., Liu, S., Zhao, H., Jia, J.: Distilling knowledge via knowledge review. In: Proceedings of the IEEE/CVF Conference on Computer Vision and Pattern Recognition. pp. 5008–5017 (2021)

[2] Jin, Y., Wang, J., Lin, D.: Multi-level logit distillation. In: Proceedings of the IEEE/CVF Conference on Computer Vision and Pattern Recognition. pp. 24276–24285 (2023)

[3] Lin, S., Xie, H., Wang, B., Yu, K., Chang, X., Liang, X., Wang, G.: Knowledge distillation via the target-aware transformer. In: Proceedings of the IEEE/CVF Conference on Computer Vision and Pattern Recognition. pp. 10915–10924 (2022)

[4] Miles, R., Mikolajczyk, K.: Understanding the role of the projector in knowledge distillation. In: Proceedings of the 38th AAAI Conference on Artificial Intelligence (AAAI-24)

---

> ### Author Response · Authors · 2025-06-19
>
> We express our heartfelt gratitude to the reviewer for their dedicated time and thorough evaluation of our paper. The valuable insights they offered are highly appreciated, and we are grateful for the constructive feedback. We are devoted to addressing the reviewer's concerns and integrating their suggestions into the final version to elevate the overall quality of our paper. The following section presents comprehensive responses to the reviewer's comments, queries, and recommendations to enhance the paper's overall rating.
>
> ``Ablation on the impact of C-Net``
>
> AMAL[1] maintains ‘n’ i.e. instance  number of learnable parameters. In the below table we present a comparison between  AMAL, MetaDistil and MC-Distil. In the paper we include only MetaDistil as it seemed a more fairer baseline to compare MC-Distil. We observe that while AMAL slightly outperforms MetaDistil, it is consistently outperformed by MC-Distil. As outlined in the baseline section (section 4.2), MetaDistil shares the same objective as AMAL but replaces the direct estimation of instance-wise mixing parameters with a parameter-efficient C-Net, which approximates these weights. This approximation leads to a marginal drop in performance for MetaDistil compared to AMAL. In contrast, MC-Distil leverages the shared C-Net not only for parameter efficiency but also as a coordination mechanism that captures the relative learning dynamics across a cohort of students with varying capacities. This enables more informed and effective distillation, resulting in superior performance across all student models.
>
> | Teacher      | Teacher Acc. | Student     | CE    | KD    | AMAL  | MetaDistil | MC-DISTIL  |
> |--------------|--------------|-------------|-------|-------|-------|-------------|--------|
> |              |               | ResNet10-xxs| 31.85 | 33.45 | 34.98 | 34.92       | **36.19** |
> | ResNet10-l   | 72.2         | ResNet10-xs | 42.75 | 44.87 | 46.69 | 46.01       | **47.57** |
> |              |              | ResNet10-s  | 52.48 | 55.38 | 57.15 | 57.20       | **58.36** |
> |              |              | ResNet10-m  | 64.28 | 66.93 | 68.60 | 68.28       | **69.42** |
> |--------------|--------------|-------------|-------|-------|-------|-------------|--------|
> |              |              |ResNet10-xxs| 31.85 | 33.95 | 34.93 | 34.66       | **35.97** |
> | ResNet10     | 75.18        | ResNet10-xs | 42.75 | 44.87 | 46.93 | 46.32       | **47.53** |
> |              |              | ResNet10-s  | 52.48 | 55.56 | 57.90 | 57.78       | **58.10** |
> |              |              | ResNet10-m  | 64.28 | 67.27 | 68.90 | 68.89       | **69.21** |
> |--------------|--------------|-------------|-------|-------|-------|-------------|--------|
> |              |              | ResNet10-xxs| 31.85 | 33.56 | 34.60 | 34.40       | **35.94** |
> | ResNet18     | 76.99        | ResNet10-xs | 42.75 | 45.02 | 46.63 | 46.24       | **46.99** |
> |              |              | ResNet10-s  | 52.48 | 55.73 | 57.43 | 57.40       | **57.50** |
> |              |              | ResNet10-m  | 64.28 | 66.42 | 68.43 | 68.43       | **68.45** |
> |--------------|--------------|-------------|-------|-------|-------|-------------|--------|
> |              |              | ResNet10-xxs| 31.85 | 33.32 | 33.94 | 33.76       | **36.10** |
> | ResNet34     | 79.47        | ResNet10-xs | 42.75 | 44.94 | 46.89 | 46.43       | **47.12** |
> |              |              | ResNet10-s  | 52.48 | 54.73 | 57.21 | 56.91       | **57.67** |
> |              |              | ResNet10-m  | 64.28 | 66.52 | 68.11 | 68.09       | **68.24** |
> |--------------|--------------|-------------|-------|-------|-------|-------------|--------|
>
>
>
> Note that both AMAL and MetaDistil represent the case of “if one was to use independent learnable loss weights for each student individually”. In AMAL paper authors have shown that that such dynamic, instance-specific weighting strategies outperform static or scalar mixing approaches.
>
> Also, `weights proportional to the capacity of the student` is equivalent to static or scalar mixing approaches as our method doesn’t propose a multi-student framework for KD; instead, we use a set of student model cohorts only during the training phase to improve the efficacy of the KD in situations where there is a vast gap between the learning capacity of the teacher and the student models. During inference, none of the additional models are present. Once trained during inference, our method has no extra cost over traditional KD.
>
>
> [1] D. Sivasubramanian, A. Maheshwari, P. AP, P. Shenoy, and G. Ramakrishnan, “Adaptive Mixing of Auxiliary Losses in Supervised Learning”, AAAI, vol. 37, no. 8, pp. 9855-9863, Jun. 2023.

---

> > ### Author Response · Authors · 2025-06-19
> >
> > ``Results on MS COCO dataset for object detection``
> >
> > We evaluate the effectiveness of MC-Distil for student-teacher distillation in object detection using the Fast R-CNN framework on a standard benchmark. A ResNet-50 serves as the teacher, while ResNet-18 and ResNet-34 are used as student models. As shown in the tables, MC-Distil consistently outperforms both the baseline (student trained from scratch) and standard knowledge distillation (KD) across all metrics. For instance, with ResNet-18 as the student, MC-Distil improves the AP from 28.04 to 38.20, and AP75 from 28.90 to 41.59 — indicating substantial gains in precise localization. While standard KD yields modest improvements (e.g., AP75: 32.86), MC-Distil provides a significantly larger boost, especially at stricter thresholds like AP75, where fine-grained box alignment is critical. Similar trends are observed for ResNet-34, where MC-Distil achieves 46.48 AP75, up from 38.78 with KD. These results highlight that MC-Distil is particularly effective in transferring detailed spatial knowledge in the Fast R-CNN setting, enabling lightweight students to better mimic the localization capabilities of deeper teacher models.
> >
> > AP
> > | Teacher   | Teacher Accuracy | Student   | CE | KD    | MC-Distil |
> > |-----------|------------------|-----------|------------------|-------|-----------|
> > | ResNet50  | 46.10            | ResNet18  | 28.04            | 31.49 | 38.20     |
> > |           |                  | ResNet34  | 31.10            | 37.11 | 42.70     |
> >
> > AP50
> > | Teacher   | Teacher Accuracy | Student   | CE | KD    | MC-Distil |
> > |-----------|------------------|-----------|------------------|-------|-----------|
> > | ResNet50  | 60.00            | ResNet18  | 45.45            | 51.12 | 52.90     |
> > |           |                  | ResNet34  | 51.71            | 57.41 | 58.49     |
> >
> > AP75
> > | Teacher   | Teacher Accuracy | Student   |CE  | KD    | MC-Distil |
> > |-----------|------------------|-----------|------------------|-------|-----------|
> > | ResNet50  | 50.20            | ResNet18  | 28.90            | 32.86 | 41.59     |
> > |           |                  | ResNet34  | 32.34            | 38.78 | 46.48     |

---

### Review · Reviewer_x9rt · 2025-06-06

**Summary Of Contributions:**

The paper introduces "Meta-Collaborative Distillation" (MC-Distil), a novel approach to knowledge distillation (KD) where multiple student models of varying capacities collaboratively learn from a single teacher model. A coordinator network (C-NET) facilitates this by adaptively weighting training losses based on instance hardness and student capacity, leading to improved accuracy across diverse datasets (e.g., CIFAR100, Tiny-ImageNet, ImageNet). Key contributions include:

1. Enhanced student model performance, with average accuracy gains of 2.5% on CIFAR100 and 2% on Tiny-ImageNet.
2. Novel insight that larger students benefit from collaboration with smaller ones, a departure from traditional multi-teacher approaches.
3. Robustness to real-world challenges like label noise and domain adaptation, outperforming state-of-the-art baselines (e.g., TAKD, DGKD, SHAKE).

**Audience:**

Yes

**Claims And Evidence:**

Yes

**Requested Changes:**

1. Quantify computational costs (e.g., training time, memory usage) of MC-Distil compared to baselines to clarify trade-offs.
2.  Expand evaluation to real-world datasets or applications to validate practical utility.

**Strengths And Weaknesses:**

Strengths:
1. Innovative use of C-NET for meta-collaboration, effectively addressing capacity gaps in KD.
2. Comprehensive experiments across varied architectures, datasets, and scenarios, with statistically significant gains.
3. Demonstrates robustness in extreme teacher-student gaps and noisy data settings.

Weaknesses:
1. Limited discussion on computational trade-offs of bi-level optimization and C-NET training.
2. Proof-of-concept focused on benchmark datasets; real-world applicability remains untested.
3. Scalability of MC-Distil with very large student cohorts or datasets is underexplored.

---

> ### Author Response · Authors · 2025-06-19
>
> We express our heartfelt gratitude to the reviewer for their dedicated time and thorough evaluation of our paper. The valuable insights they offered are highly appreciated, and we are grateful for the constructive feedback. We are devoted to addressing the reviewer's concerns and integrating their suggestions into the final version to elevate the overall quality of our paper. The following section presents comprehensive responses to the reviewer's comments, queries, and recommendations to enhance the paper's overall rating.
>
> ``Quantify computational costs``
> Compared to the baselines such as TAKD, DGKD and DML, which use multiple student models during training, MC-Distil additionally introduces a C-Net. For a student pool of  ResNet10-xxs, ResNet10-xs, ResNet10-s, and ResNet10-m, ResNet34 as the teacher model, and ResNet10-l as the C-Net we incur just 5% of additional FLOPS. Further, since the reduction in C-Net size doesn’t affect the MC-Distil performance drastically (as shown in appendix C.2) the C-Net size could be further optimized.
> Both C-Net and student cohorts are presented only during the training phase. Our method doesn’t require any ensemble framework or multistudent framework at the time of inference; therefore, our method has no additional cost compared to traditional KD during inference.
>
> ``real-world applicability remains untested``
> To show real world relevance, apart from CIFAR100 and TinyImageNet datasets we also perform experiments with two other datasets with instance dependent label noise viz. instance CIFAR-100 and Clothing-1M (section 4.7). Here we show that MC-Distill outperforms most SOTA of the noise reducing methods. We also demonstrate the models trained with MC-Distill have better generalisability(section 4.6). We report the performance obtained on TinyImageNet-C  using the model trained with the TinyImageNet dataset(Table 6). We report performance on various levels of corruption. Models trained with MC-Distill perform best on corrupted TinyImageNet-C dataset.  Also, we represent results on iWildCam dataset which poses a significant challenge due to alterations in the capturing environment, resulting in a demanding test set in Table 4.
>
> ``Scalability of MC-Distil with very large student cohorts or datasets is underexplored``
> To further demonstrate scalability, we extend this analysis to six student models of increasing capacity on the CIFAR-100 dataset. As shown in the table below, MC-Distil consistently outperforms both cross-entropy (CE) training and standard knowledge distillation (KD) across all student sizes — from very compact models like ResNet10-xxs to full-capacity ResNet10. For instance, the smallest model (ResNet10-xxs) improves from 31.85% (CE) to 35.94% (MC-Distil), while even the largest student (ResNet10) benefits, reaching 77.2% accuracy compared to 76.73% with KD. This consistent improvement highlights MC-Distil's robustness across a wide range of model capacities.
>
> | Teacher   | Teacher Acc | Student      | CE    | KD    | MC-Distil |
> |-----------|-------------|--------------|-------|-------|----------|
> |           |             | ResNet10-xxs | 31.85 | 33.56 | **35.94** |
> |           |             | ResNet10-xs  | 42.75 | 45.02 | **46.99** |
> | ResNet18  | 76.99       | ResNet10-s   | 52.48 | 55.73 | **57.5**  |
> |           |             | ResNet10-m   | 64.28 | 66.42 | **68.45** |
> |           |             | ResNet10-l   | 72.2  | 73.85 | **74.55** |
> |           |             | ResNet10     | 75.18 | 76.73 | **77.2**  |
>
> Additionally, as demonstrated in the main paper, MC-Distil scales effectively to ImageNet, a large-scale dataset with over 1.2 million training images and 1,000 classes. This further reinforces the method’s applicability in real-world scenarios involving high-volume data and diverse model cohorts, confirming its utility for both low-resource and large-scale settings.

---

> ### Author Response · Authors · 2025-06-22
> **Computational Cost and Time-Based Analysis**
>
> We sincerely thank the reviewer for suggesting a computational and time-based analysis. This was very helpful, and we believe the added comparison strengthens our work by highlighting the efficiency of our method.
>
> As recommended, we profiled several distillation techniques. The table below summarizes the total computational cost (in TFLOPs) and training time required to train all student models for one epoch.
>
> | Method            | GFLOPs per Batch | Total TFLOPs per Epoch | Per-Epoch Training Time (s) |
> |-------------------|------------------|-------------------------|-----------------------------|
> | **TAKD [1]**      | 1.49             | 0.58                    | 20.06                       |
> | **DGKD [2]**      | 4.77             | 1.86                    | 24.63                       |
> | **RMC [3]**       | 4.74             | 1.85                    | 12.49                       |
> | **SHAKE [4]**     | 18.63            | 7.26                    | 14.39                       |
> | **MetaDistill [5]** | 10.29          | 4.01                    | 9.10                        |
> | **MC-Distil**     | 2.47             | 0.97                    | 11.90                       |
>
> ### Key Observations
>
> - **Sequential Bottleneck:** Methods such as **TAKD** [1] and **DGKD** [2] require the sequential training of progressively smaller models. Consequently, their total training time is the sum of the times for each stage, making them significantly slower at producing the full cohort of students compared to our simultaneous approach.
>
> - **Auxiliary Model Overhead:** Methods like **SHAKE** [4] and **MetaDistill** [5], which achieve per-epoch training times comparable to **MC-Distil**, rely on auxiliary models that must be trained separately for each student. For example, SHAKE [4] requires a distinct proxy teacher for each student, while MetaDistill [5] uses a separate meta-network for loss re-weighting. To train the same number of students as our model, these approaches must be executed independently for each student, making them more computationally expensive in terms of total required resources and FLOPs.
>
> - **Accuracy vs. Efficiency:** While methods like **RMC** [3] are computationally lighter than some of the more complex meta-learning strategies, our proposed **MC-Distil** significantly outperforms it in terms of final model accuracy, as demonstrated in the table below. This highlights our method’s ability to achieve a superior balance of efficiency and performance.
>
> ### Accuracy Comparison with ResNet34 Teacher
>
> To further contextualize these findings, the following table presents a direct accuracy comparison between **RMC** [3] and **MC-Distil** on CIFAR-100 and Tiny-ImageNet, using ResNet34 as the teacher. This setup corresponds to the computational analysis described above (Further details about setup are in the next section).
>
> | Dataset         | Student Model | RMC      | **MC-Distil (Ours)** |
> |------------------|----------------|----------|-----------------------|
> | **CIFAR-100**    | ResNet10-xxs   | 34.46%   | **36.10%**            |
> |                  | ResNet10-xs    | 42.78%   | **47.12%**            |
> |                  | ResNet10-s     | 53.58%   | **57.67%**            |
> |                  | ResNet10-m     | 65.58%   | **68.24%**            |
> | **Tiny-ImageNet**| ResNet10-xxs   | 13.78%   | **15.24%**            |
> |                  | ResNet10-xs    | 19.28%   | **21.82%**            |
> |                  | ResNet10-s     | 25.99%   | **29.59%**            |
> |                  | ResNet10-m     | 33.58%   | **38.34%**            |
>
> This analysis underscores that our method not only offers significant computational advantages but also consistently achieves higher accuracy across all student models and datasets in this setting.

---

> > ### Author Response · Authors · 2025-06-22
> > **Computational Cost and Time-Based Analysis**
> >
> > ### Calculation Methodology
> >
> > The values presented in the table are based on profiling the run for one epoch, and are designed to provide a fair comparison of the total computational cost required to train a full cohort of student models. The following principles were applied:
> >
> > 1. A standard heuristic was used to estimate the computational cost of network operations. A model's forward pass contributes `1x` its theoretical FLOPs. A full training step, comprising both a forward and a backward pass for gradient computation, is estimated to contribute `3x` its forward pass FLOPs [6].
> >
> > 2. The experimental setup profiles the total time and workload required to train the entire student cohort, which consists of `ResNet10_m`, `ResNet10_s`, `ResNet10_xs`, and `ResNet10_xxs`. The primary teacher model is a ResNet34 architecture. All reported metrics—both time and FLOPs—are normalized to a per-epoch basis. The auxiliary models are specified as follows: the proxy teacher in **SHAKE** [4] is a `ResNet18`; the meta-model in **MetaDistill** [5] (similar to AMAL [5]) and the C-Net in **MC-Distil** are both `ResNet32`; and the cohort of student models is used for difficulty estimation in **RMC** [3].
> >
> > 3. For sequential distillation methods such as **TAKD** [1] (Teacher → TA₁ → TA₂ ...) and **DGKD** [2] (Teacher + TAs → next TA), the total per-epoch time and FLOPs are calculated as the **sum of the costs of each required stage**. This cumulative value represents the total workload to complete one epoch of training for every model in the sequential chain.
> >
> > 4. For methods that require independent training sessions for each student, such as **SHAKE** [4], **MetaDistill** [5], and **RMC** [3], the following approach was taken to represent total workload versus ideal parallel execution:
> >    - The **Total FLOPs** values are calculated as the **sum of the computational costs for all required independent runs** needed to train the entire student cohort. This provides a true measure of the total computational budget.
> >    - The **Total Time** reported is the profiled wall-clock time for only the **most computationally expensive run** (i.e., the one involving `ResNet10_m`). This metric represents the best-case parallel execution time, assuming sufficient hardware to run all student trainings concurrently.
> >
> > ### References
> >
> > [1] Mirzadeh, S. I. et al. (2020). *Improved knowledge distillation via teacher assistant.* AAAI 2020.
> > [2] Son, W. et al. (2021). *Densely guided knowledge distillation using multiple teacher assistants.* ICCV 2021.
> > [3] Du, M. et al. (2023). *Robustness challenges in model distillation and pruning for natural language understanding.* EACL 2023.
> > [4] Li, L. & Jin, Z. (2022). *Shadow knowledge distillation: Bridging offline and online knowledge transfer.* NeurIPS 2022.
> > [5] Sivasubramanian, D. et al. (2023). *Adaptive mixing of auxiliary losses in supervised learning.* AAAI 2023.
> > [6] Zhou, X. et al. (2021). Efficient neural network training via forward and backward propagation sparsification. NeurIPS 2021 (arXiv:2111.05685).

---

### Decision · Action_Editor_4pzM · 2025-07-08

**Recommendation:** Accept as is

**Audience:**

Yes

**Audience Explanation:**

Knowledge distillation is a popular technique in ML community.

**Claims And Evidence:**

Yes

**Claims Explanation:**

This paper introduces Meta-Collaborative Distillation (MC-DISTIL), a knowledge distillation framework designed to address distillation when a significant capacity gap exists between teacher and student models. The key idea is to facilitate collaborative learning among a cohort of student models with varying capacities, all learning from a single teacher. This collaboration is orchestrated by a Coordinator Network (C-NET), which functions as a meta-learner. C-NET modulates the training loss for each student on an instance-by-instance basis by reweighting the standard cross-entropy loss and the teacher-matching KL divergence loss. The C-NET learns from each student's performance and training instance characteristics, enabling students of different capacities to improve together.

The authors demonstrate that MC-DISTIL achieves significant accuracy gains on datasets like CIFAR100, TinyImageNet, and ImageNet on multiple student and teacher architectures, outperforming existing state-of-the-art methods. During the rebuttal period, detection task was also included.

After the rebuttal, all three reviewers are positive on the submission.